# communications

# biology

# Developmental timing-dependent organization of synaptic connections between mossy fibers and granule cells in the cerebellum

Taegon Kim [1], Heeyoun Park [1], Keiko Tanaka-Yamamoto [1✉] & Yukio Yamamoto [1✉]

The long-standing hypothesis that synapses between mossy fibers (MFs) and cerebellar granule cells (GCs) are organized according to the origins of MFs and locations of GC axons, parallel fibers (PFs), is supported by recent findings. However, the mechanisms of such organized synaptic connections remain unknown. Here, using our technique that enabled PF location-dependent labeling of GCs in mice, we confirmed that synaptic connections of GCs with specific MFs originating from the pontine nucleus (PN-MFs) and dorsal column nuclei (DCoN-MFs) were gently but differentially organized according to their PF locations. We then found that overall MF-GC synaptic connectivity was biased in a way that dendrites of GCs having nearby PFs tended to connect with the same MF terminals, implying that the MF origin- and PF location-dependent organization is associated with the overall biased MF-GC synaptic connectivity. Furthermore, the development of PN-MFs preceded that of DCoN-MFs, which matches the developmental sequence of GCs that preferentially connect with each type of these MFs. Thus, our results revealed that overall MF-GC synaptic connectivity is biased in terms of PF locations, and suggested that such connectivity is likely the result of synaptic formation between developmental timing-matched partners.

[1] Brain Science Institute, Korea Institute of Science and Technology (KIST), Seoul 02792, Republic of Korea. ✉email: keikoyamat@gmail.com; yukio.kist@gmail.com

Cerebellar granule cells (GCs) are the most numerous neurons in the brain, accounting for half of all neurons[1]. While GC somas are located in the cerebellar input layer, also called the GC layer (GCL), they extend their axons, parallel fibers (PFs), in the molecular layer (ML), and individual PFs run parallel with layer structures, resulting in the formation of synaptic contacts onto distinct regions of postsynaptic Purkinje cells (PCs) or ML interneurons. In the GCL, each GC receives inputs through on average only four short dendrites, which make synapses with mossy fiber (MF) terminals at the glomeruli. A glomerulus includes a single MF terminal and 8–20 GC dendrites[2]. There has been a traditional theory that massive numbers of GCs are advantageous for sparse coding to separate different contexts and achieve associative learning at the synapses with PC dendrites[3,4]. A computational study combined with analyses of synaptic connectivity between MFs and GC dendrites in the glomeruli demonstrated that the synaptic connectivity is beneficial for effective sparse coding[2]. Thus, the characteristics of MF-GC network structure may be utilized for their functions.

In addition to the network topology, another important characteristic of GCs is the spatial structure, which includes the fact that PFs are located at a given sublayer of the ML and consequently form the laminar structure. Such spatial organization has naturally drawn attention to its relationship with the network structure. Early studies using Golgi staining suggested an association between the locations of GC somas in the GCL and the locations of their PFs in the ML, and projections of MFs originating from specific nuclei to specific sublayers of the GCL[5,6]. Based on these studies, it was hypothesized that signals arising from MFs of different origins may be conveyed differently to the ML through sublayers of activated PFs[7]. A recent study suggested the differences in GC electrophysiological properties according to their PF locations, based on the correlation between PF locations and GC soma locations that was suggested using sparse GC labeling[8]. However, other studies[9–11], including our previous study[12], demonstrated that there is no apparent correlation between GC soma locations and their PF locations along the anterior-posterior axis. Thus, such association between synaptic organizations from MFs to GCs in the GCL and sublayers of PF projections remained under debate. This has been difficult to investigate, presumably owing to the very large numbers of GCs in a relatively small volume, which prevents systematic dissection according to the PF projections. Another recent study finally developed a genetic strategy that enabled to distinguish early- and late-born GCs, which have PFs at deep (closer to PC somas) and superficial sublayers of the ML, respectively, and analyzed the properties of these GCs[13]. By combining this strategy with retrograde monosynaptic rabies virus tracing, this recent study demonstrated different patterns of MF inputs between early- and late-born GCs. However, it remains unclear as to how such different patterns are created.

In the present study, we took advantage of the characteristic features of an adeno-associated viral (AAV) vector with the minimum region of GABAA receptor α6 subunit (GABRα6) promoter that we previously developed[14]. Using stereotaxic injection of this vector, AAV-GABRα6, expressing fluorescent proteins during postnatal development, we were able to distinctly label a specific group of GCs that have PFs located at a certain sublayer of the ML[12,15]. Our imaging analyses using this labeling and hypothetical network model demonstrated the overall organization of synaptic connectivity from MFs to GCs based on their PF locations in the ML. In addition, analyses of MF terminals during postnatal development suggested the contribution of developmental processes to the organization.

## Results

### Injection time-dependent patterns of GFP expression in GCs and PFs driven by the AAV-GABRα6.
We have previously demonstrated that stereotaxic injection of AAV-GABRα6 at a proper timing during postnatal development is capable of triggering selective molecular expression in a group of GCs, whose PFs are bundled together at approximately one third of the ML[12,15]. We confirmed the expression patterns by injecting AAV-GABRα6-GFP into lobule IV/V of the cerebellum at different postnatal times (P7–P13) and observing GFP expression at P21–P23 (Supplementary Fig. 1a), which is approximately the time when cerebellar development is completed in mice[16]. As expected from our previous studies, GFP-positive PF bundles were located at deeper or more superficial layers when stereotaxic injections of mice were performed at earlier or later ages, respectively (Fig. 1a and Supplementary Fig. 1b). Consequently, the distance between PC somas and GFP-positive PF bundles in the ML was longer in mice injected at later times (Fig. 1b and Supplementary Fig. 1c). Because the rise time of excitatory postsynaptic potentials or currents (EPSCs) recorded from PC somas is dependent on the location of synaptic stimulation due to dendritic filtering[8,17], we recorded EPSCs from PC somas by stimulating PFs with electrodes placed around sublayers that were labeled by the P7 (P7-PFs) or P12 (P12-PFs) injection (Supplementary Fig. 1d). While the decay times were not significantly different, EPSCs elicited by P7-PF stimulation had a shorter rise time than those elicited by P12-PF stimulation (Supplementary Fig. 1e; rise time, $p < 10^{-6}$; decay time, $p = 0.202$; paired Student's $t$-test), confirming the shorter distance between PC somas and P7-PFs. In the GFP-positive areas of the ML, GFP signals overlapped with most signals of vesicular glutamate transporter 1 (vGLUT1) (Supplementary Fig. 2a), indicating nearly exhaustive labeling of PFs in the area and their GCs by GFP. Considering that PFs are made in order from the deep layer to the superficial layer of the ML during postnatal development[18], the injection time-dependent labeling of PF bundles implies that AAV-GABRα6 triggers the target molecule expression at a specific developmental stage of GCs. To confirm it, GC precursors at nearly final stage of mitosis were labeled by intraperitoneal administration of 5-ethynyl-2'-deoxyuridine (EdU) at P7, P8, P10, or P12, and this labeling was combined with stereotaxic injection of AAV-GABRα6-GFP at P10 (Supplementary Fig. 1a). GCs labeled by both EdU and GFP were often observed when EdU was administered at P7 or P8, but hardly found when EdU was administered at P12 (Supplementary Fig. 2b, c), indicating that AAV-GABRα6 triggered molecular expression mostly in postmitotic GCs at the time of injection. When AAV-GABRα6-GFP was injected at P22, GFP-positive GCs were hardly detected (Supplementary Fig. 2d), indicating that AAV-GABRα6 did not trigger expression in mature GCs at the time of injection. Previous studies revealed that the endogenous GABRα6 promoter was active in mature GCs but not in immature GCs[19–21]. We also confirmed by western blot and immunohistochemical analyses that the expression of endogenous GABRα6 protein gradually increased during postnatal development and was found only in the GCL of developing and adult mice (Supplementary Fig. 2e, f), indicating its expression in mature GCs. Unlike the endogenous GABRα6 expression pattern, our results demonstrated that exogenously introduced AAV-GABRα6 triggered sufficient molecular expression in postmitotic immature GCs at the time of injection. This property of AAV-GABRα6 enables us to systematically label only a particular group of PFs and their GCs. In addition, our labeling results confirmed the orderly differentiation of GCs[18]; earlier-born GCs have PFs in the deeper layer, whereas later-born GCs have PFs in the more superficial layer. For convenience, we refer to GCs having PFs at the deep (D), middle (M),

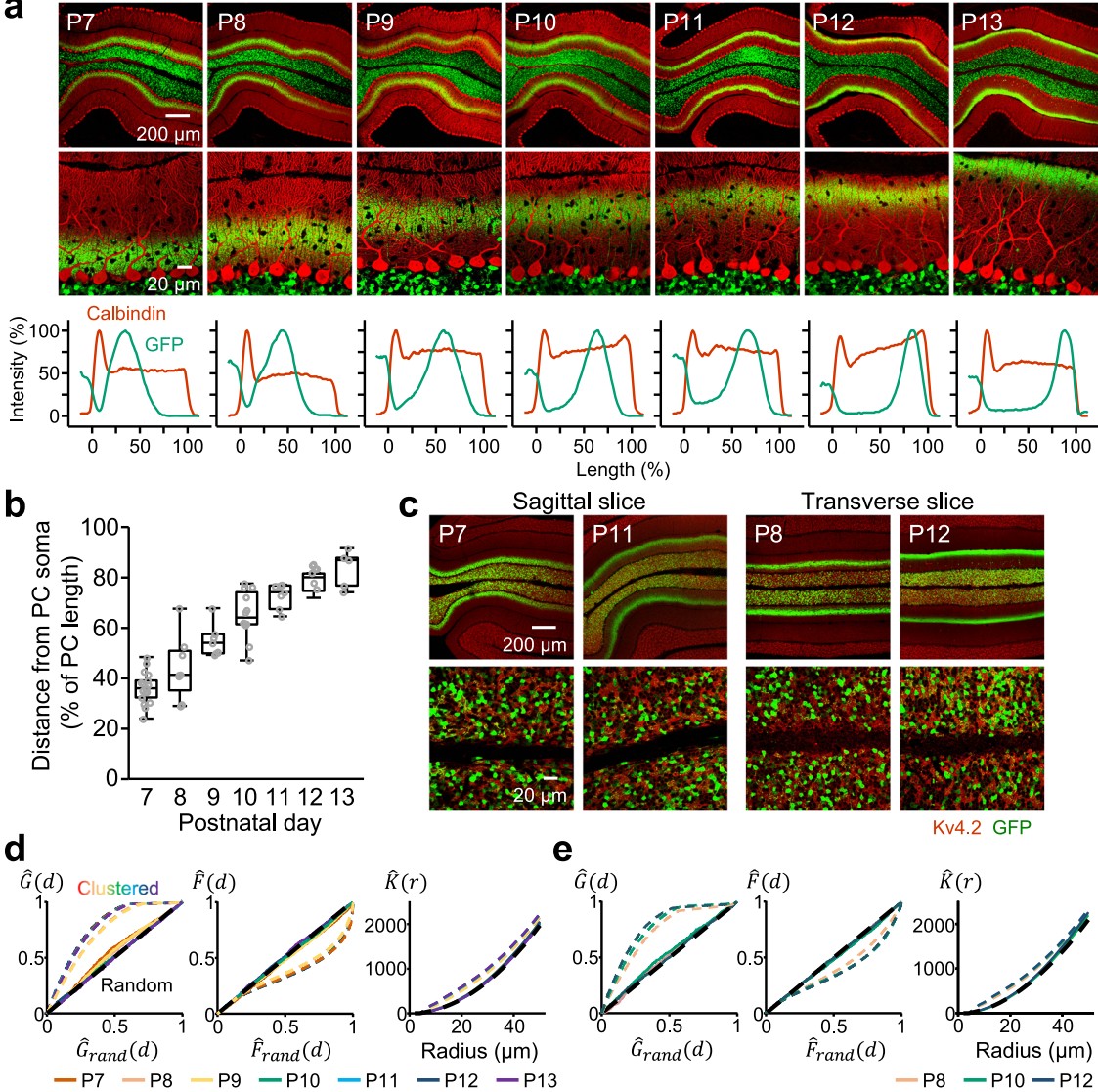

**Fig. 1 Characterization of expression patterns triggered by the injection of AAV-GABRα6-GFP into the cerebellum. a** Confocal images of lobule IV/V in cerebellar sagittal slices, where AAV was injected. Expression of GFP (green) was triggered by the injection of AAV-GABRα6-GFP on the indicated postnatal days. PCs are stained with a calbindin antibody (red). Bottom graphs show line-scan profiles across the PC layer and the ML. The x-axis indicates the length (shown as a percentage) from PC soma (0) to the edge of PC dendrites (100). **b** Distances between the labeled PF bundles and PC somas ($N = 5$–17 mice each, 63 mice in total). The center of the PF bundles was detected by fitting the line scan profiles of GFP signals (see in **a**) to the Gaussian function. Data are presented as boxplots. **c** Confocal images of sagittal (left) or transverse (right) slices stained with a Kv4.2 antibody (red). AAV-GABRα6-GFP (green) was injected on the indicated postnatal days. **d**, **e** The analyses of GC distributions in sagittal (**d**) or transverse (**e**) planes by the $\hat{G}$ (left), $\hat{F}$ (middle), or $\hat{K}$ functions (right) ($N = 3$–5 mice, 50 mice in total, 3 images per mouse). Black and colored dotted lines show the results obtained from computationally created random and clustered configurations, respectively. Solid lines show the experimental results.

and superficial (S) sublayers, as D-GCs, M-GCs, and S-GCs, respectively. D-GCs mainly consist of GCs labeled by P7 or P8 injection, M-GCs mainly consist of GCs labeled by P9, P10, or P11 injection, and S-GCs mainly consist of GCs labeled by P12 or P13 injection.

Our exhaustive labeling of GCs that have PFs in a given sublayer of the ML seemed to be an appropriate technique to interrogate the spatial organization of GC somas and their PFs. We thus analyzed labeled GCs in sagittal and transverse slices, to determine whether GC soma locations are organized along the anterior-posterior, dorsal-ventral, or lateral axis according to the positions of their PFs. In both sagittal and transverse slices, GFP-positive GC somas appeared to be clustered neither in the perpendicular nor parallel direction to the layers but were dispersed throughout the GCL, upon the injection at any age (Fig. 1c). The random and clustered

configurations of GC somas were computationally generated (Supplementary Fig. 3a, b, see Methods), to compare distributions with experimental results using first order nearest neighbor distance statistics, namely G and F functions (Supplementary Fig. 4a–c, see Methods). The distributions of GCs labeled at any age were similar to the random configurations rather than the clustered patterns (Fig. 1d, e, left and middle). We then used the well-known second order statistic, Ripley's K function (Fig. 1d, e, right), which also showed that the estimated K functions from the experimental results are equivalent to those in the random configurations. Thus, the G, F, and Ripley's K functions confirmed that GFP-positive GC somas are randomly distributed along the anterior-posterior, dorsal-ventral, or lateral axis at any time of injection. These results indicate that PF location does not correlate with GC soma location along any axis.

**Synaptic connections with MF terminals originating from specific nuclei**. A recent study demonstrated that sensory and motor nuclei preferentially provide MF inputs to early-born GCs, whereas basal pontine and deep cerebellar nuclei preferentially provide MF inputs to late-born GCs[13]. The analyses were performed in lobule VI using labeling of early- and late-born GCs with retrograde rabies virus tracing. In the present study, we tested whether such biased synaptic connections can also be detected in lobule IV/V using our labeling method that is capable of differential labeling of D-GCs and S-GCs by injecting AAV-GABRα6-tdTomato (tdT) at P7 and AAV-GABRα6-GFP at P12 (Fig. 2a). As shown in the schematic drawing of Fig. 2b, terminals of MFs preferentially providing inputs to S-GCs would include more GFP signals than tdT signals, and vice versa. Because both MFs originating from the pontine nucleus (PN-MFs) and dorsal column nuclei (DCoN-MFs) have been shown to project to lobule IV/V[22], we analyzed GFP and tdT signals in these MF terminals that were visualized by injecting AAV vectors with a short fragment of the synapsin-1 gene promoter expressing mTagBFP2 (AAV-sSyn-mTagBFP) into the PN or the DCoN (Fig. 2c, d). The normalized intensity of GFP ($D_{GFP}$) was significantly higher than the normalized intensity of tdT ($D_{tdT}$) in both PN-MFs and DCoN-MFs (Fig. 2e; PN-MF, $p = 3.8 \times 10^{-4}$; DCoN-MF, $p = 1.2 \times 10^{-9}$; one-way ANOVA followed by the Bonferroni test), suggesting that both PN-MFs and DCoN-MFs connect to S-GCs more than D-GCs. One may wonder if the difference in $D_{GFP}$ and $D_{tdT}$ would be due to the difference in properties of fluorescent molecules, e.g., easier transportation of GFP than tdT. However, when we analyzed $D_{GFP}$ and $D_{tdT}$ in non-specific glomeruli that were visualized by triple staining with antibodies against vGLUT1, vesicular GABA transporter (vGAT), and Kv4.2 voltage-gated potassium channels (Fig. 2f), $D_{GFP}$ and $D_{tdT}$ were equivalent (Fig. 2g; $p = 0.65$; Mann–Whitney test). This indicates that the difference in $D_{GFP}$ and $D_{tdT}$ in specific MF terminals (Fig. 2e) was not due to differences in properties of fluorescent molecules, but rather to a difference in the number of GFP- and tdT-positive GC dendrites connecting with these MF terminals. In addition, both $D_{tdT}$ and $D_{GFP}$ were significantly higher in DCoN-MFs than those in PN-MFs (Fig. 2e; $D_{tdT}$, $p = 0.022$; $D_{GFP}$, $p = 4.8 \times 10^{-7}$; one-way ANOVA followed by the Bonferroni test), suggesting different aspects of synaptic connections in PN-MF terminals and DCoN-MF terminals, such as more unlabeled M-GC dendrites connecting with PN-MFs. Consistent with a previous study analyzed in lobule VI[13], our results also showed the non-uniform synaptic connections in lobule IV/V between MFs from specific origins and the dendrites of GCs having PFs at different sublayers of the ML.

**Dendrites of GCs having neighboring PFs preferably share glomeruli**. We next asked whether overall synaptic connections of GC dendrites with presynaptic MF terminals are generally organized. Three possible MF-GC network topologies can be considered, which we refer to as preferential, random, and avoidance connections (Fig. 3a). In the preferential connection, dendrites of GCs having neighboring PFs are preferentially assembled in individual glomeruli (Fig. 3a, left). On the other hand, in the avoidance connection, such assembly is avoided and consequently dendrites of GCs having dispersed PFs are assembled in individual glomeruli (Fig. 3a, right). In the random connection, there is no such organization (Fig. 3a, middle). We simply expected that these different properties could be detected by analyzing ratios of separately labeled two groups of GC dendrites in individual glomeruli. In the preferential connection, two groups of GC dendrites tend to be segregated in different glomeruli, so that the ratios are speculated to widely vary compared

with the random connection. On the other hand, in the avoidance connection, two groups of GC dendrites tend to be well mixed in individual glomeruli, so that the ratios would be relatively constant compared with the random connection. Thus, we labeled two different groups of GCs by injecting AAV-GABRα6-tdT and AAV-GABRα6-GFP at different times (Fig. 3b), and analyzed the distributions of the ratios of tdT and GFP in individual glomeruli ($R_{tdT/G} = D_{tdT}/D_{GFP}$), to distinguish three different connections. As expected, a longer interval between the two injections resulted in a longer distance between the two PF bundles (Fig. 3c, top). Although GC somas labeled with tdT and GFP were mostly segregated, some GCs were doubly labeled with both tdT and GFP, and the percentage of such overlap expression was higher in cases of shorter distances between the two PF bundles (Fig. 3c, bottom). To rule out the interference of expression of molecules by the two injections, we compared the GC fractions with GFP expression in the single injection with those in the second injection that was used to label two GC groups. The results showed no significant differences between them (Supplementary Fig. 4d; P9, $p = 0.686$; P10, $p = 0.156$; P11, $p = 0.937$; P12, $p = 0.277$; P13, $p = 0.387$; Mann–Whitney test), indicating little or no effects of the preceding injection on the molecule expression by the subsequent injection.

For the analysis of $R_{tdT/G}$, glomeruli were visualized by triple staining with antibodies against vGLUT1, vGAT, and Kv4.2, and individual glomeruli were identified by building the semi-automated segmentation procedure (Supplementary Fig. 5a, see Methods). For comparison, the ratios of putative random connections were obtained from shuffled pairs of $D_{tdT}$ and $D_{GFP}$ ($R_{tdT/G-shuffle}$, see Methods), because a random connection would be comparable to cases in which GC dendrites with tdT or GFP signals are randomly included in individual glomeruli. To compare the distributions of $R_{tdT/G}$ of experimental data ($R_{tdT/G-data}$) and $R_{tdT/G-shuffle}$, the cumulative curves were then overlaid in Fig. 3d. The distribution of $R_{tdT/G-data}$ was narrower by the P7&P9 and P10&P13 injections, and slightly narrower by the P7&P10 or P7&P11 injections, compared with $R_{tdT/G-shuffle}$ (Fig. 3d). When the distance between two PF bundles became longer by P7&P12 and P7&P13 injections, the distribution of $R_{tdT/G-data}$ became wider than $R_{tdT/G-shuffle}$ (Fig. 3d). We quantified these differences in the distribution broadness between $R_{tdT/G-data}$ and $R_{tdT/G-shuffle}$ by subtracting the estimated standard deviations of $R_{tdT/G-data}$ from those of $R_{tdT/G-shuffle}$ ($\bar{\triangle}\hat{\sigma}$, see Methods). The P7&P12 and P7&P13 injections resulted in significantly negative values of $\bar{\triangle}\hat{\sigma}$, while other times of injections resulted in significantly positive values (Fig. 3e; P7&P9, $p < 10^{-15}$; P7&P10, $p = 6.35 \times 10^{-5}$; P7&P11, $p = 9.63 \times 10^{-5}$; P7&P12, $p = 1.38 \times 10^{-6}$; P7&P13, $p = 5.18 \times 10^{-10}$; P10&P13, $p < 10^{-15}$; one-sample Wilcoxon signed rank test for comparison with 0). The results of differences in the distribution broadness confirmed the broader distributions of $R_{tdT/G-data}$ values in the case of a longer distance between two PF bundles, yet narrower distributions of $R_{tdT/G-data}$ values in the case of a shorter distance between two PF bundles, compared with $R_{tdT/G-shuffle}$. Thus, the analysis of $R_{tdT/G}$ values in individual glomeruli superficially suggests that the dendrites of D-GCs and S-GCs, with a longer distance between two PF bundles, tend to be segregated in different glomeruli, whereas dendrites of D-GCs and M-GCs or dendrites of M-GCs and S-GCs, with a shorter distance between two PF bundles, tend to co-exist in the same glomeruli.

To more properly interpret our experimental results, we developed network models of three different topologies with preferential, random, and avoidance connections. The nodes of each network model comprised 3000 GCs and 1000 glomeruli (Supplementary Fig. 6), which are comparable to their numbers

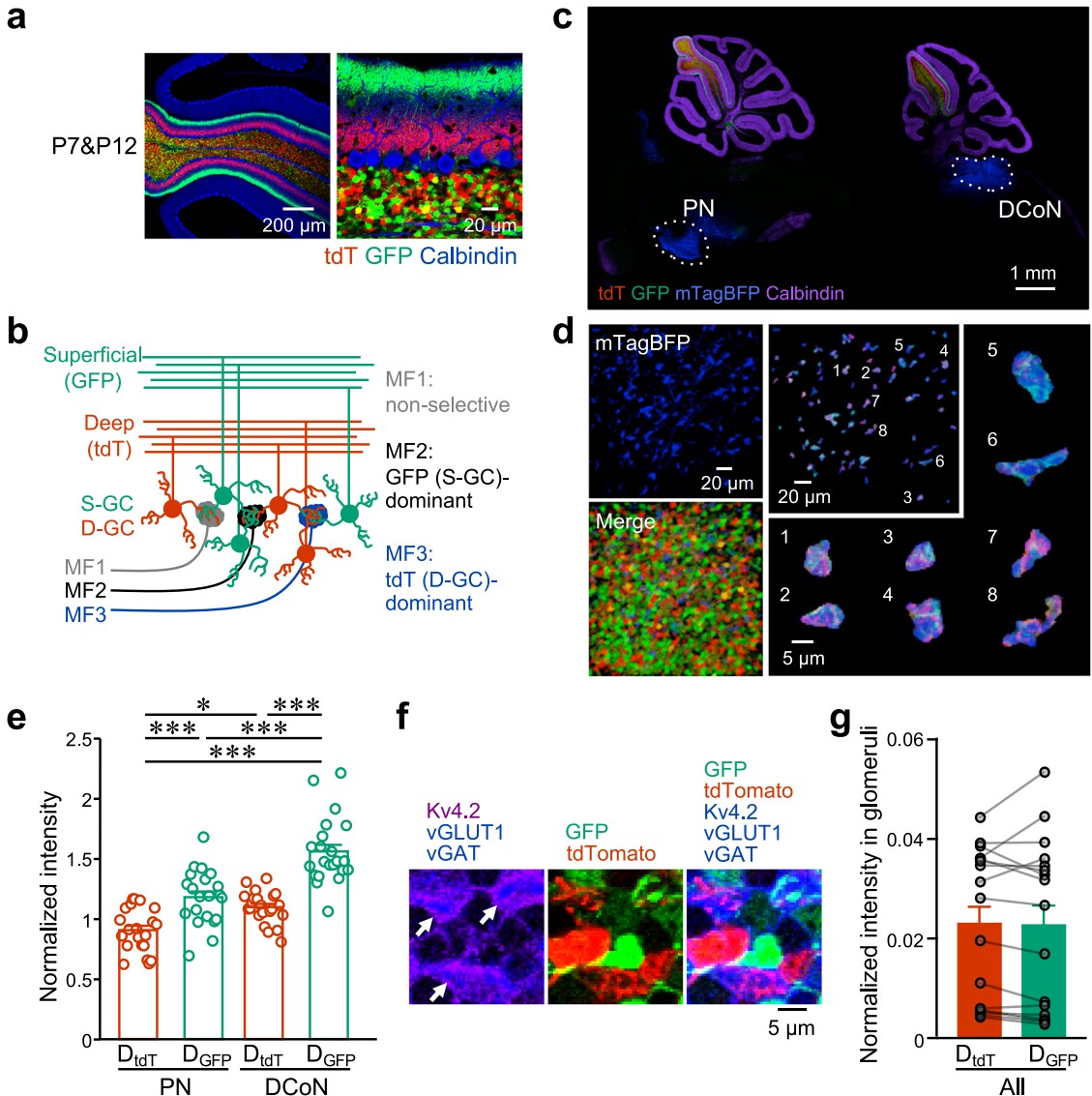

**Fig. 2 Non-uniform synaptic connections with specific MF terminals originating from the PN or the DCoN. a** Confocal images of cerebellar slices expressing tdT (red) in D-GCs and GFP (green) in S-GCs by injection of AAV-GABRα6-tdT at P7 and AAV-GABRα6-GFP at P12. Slices were stained (blue) with a calbindin antibody. **b** A schematic drawing showing synaptic connections between specific types of MFs and D-GCs or S-GCs. In the drawing, the MF2 dominantly connects with S-GCs and its terminal includes more GFP than tdT, whereas the MF3 dominantly connects with D-GCs and its terminal includes more tdT than GFP. **c** Confocal images of sagittal slices including whole cerebellum together with PN or DCoN. Slices were stained with a calbindin antibody (purple). Two groups of GCs are labeled with tdT (red) and GFP (green) by double injection at P7&P12, and MFs originating from PN (left) or DCoN (right) are labeled with mTagBFP (blue). **d** Left, representative 3D projection images of GCL, when two groups of GCs are labeled with tdT (red) and GFP (green), and DCoN-MFs are labeled with mTagBFP (blue). Right, representative 3D images of mTagBFP (blue)-positive DCoN-MF terminals, including GFP (green)-positive and tdT (red)-positive GC dendrites. **e** Averages of normalized intensities of tdT ($D_{tdT}$) and GFP ($D_{GFP}$) within the individual PN- and DCoN-MF terminals labeled with mTagBFP. Data in the individual 3D images are shown with open circles. ***$p = 3.8 \times 10^{-4}$ (PN, $D_{tdT}$ vs PN, $D_{GFP}$), ***$p = 1.2 \times 10^{-9}$ (DCoN, $D_{tdT}$ vs DCoN, $D_{GFP}$), *$p = 0.022$ (PN, $D_{tdT}$ vs DCoN, $D_{tdT}$), ***$p = 4.8 \times 10^{-7}$ (PN, $D_{GFP}$ vs DCoN, $D_{GFP}$), ***$p = 2.2 \times 10^{-15}$ (PN, $D_{tdT}$ vs DCoN, $D_{GFP}$), $p = 1$ (PN, $D_{GFP}$ vs DCoN, $D_{tdT}$), one-way ANOVA followed by the Bonferroni test ($N = 7$ mice, 14 mice in total, 3–4 images per mouse). **f** Confocal images showing tdT- (red) and GFP-positive GC dendrites (green) included in glomeruli, which are visualized by staining with three antibodies, Kv4.2 (purple in left panels or blue in right panels), vGLUT1 (blue), and vGAT (blue). Glomeruli are shown with arrows. **g** Averages of normalized intensities of tdT ($D_{tdT}$) and GFP ($D_{GFP}$), labeled by double injection at P7&P12, within non-specific glomeruli visualized by triple staining with antibodies against vGLUT1, vGAT, and Kv4.2. Gray circles represent individual data points. $p = 0.65$, Mann–Whitney test ($N = 8$ mice, 3–5 images per mouse). Data are presented as mean ± s.e.m.

in ROIs that were used for the analyses of experimental results. Individual GCs were limited to having four dendrites that connected to distinct glomeruli. The identification (ID) numbers of GCs were assigned, which reflected the developmental sequence of PFs. Therefore, the GCs with the lower ID number represent those stretching out PFs at the deeper sublayer. In the

network model with random connections, the number of GC dendrites connecting to an individual glomerulus was an average of 12 and followed a binomial distribution (Fig. 4a), which is consistent with a previous study[2]. Preferential and avoidance connections were made from random connections by the permutation of connections based on averaged distances of GC

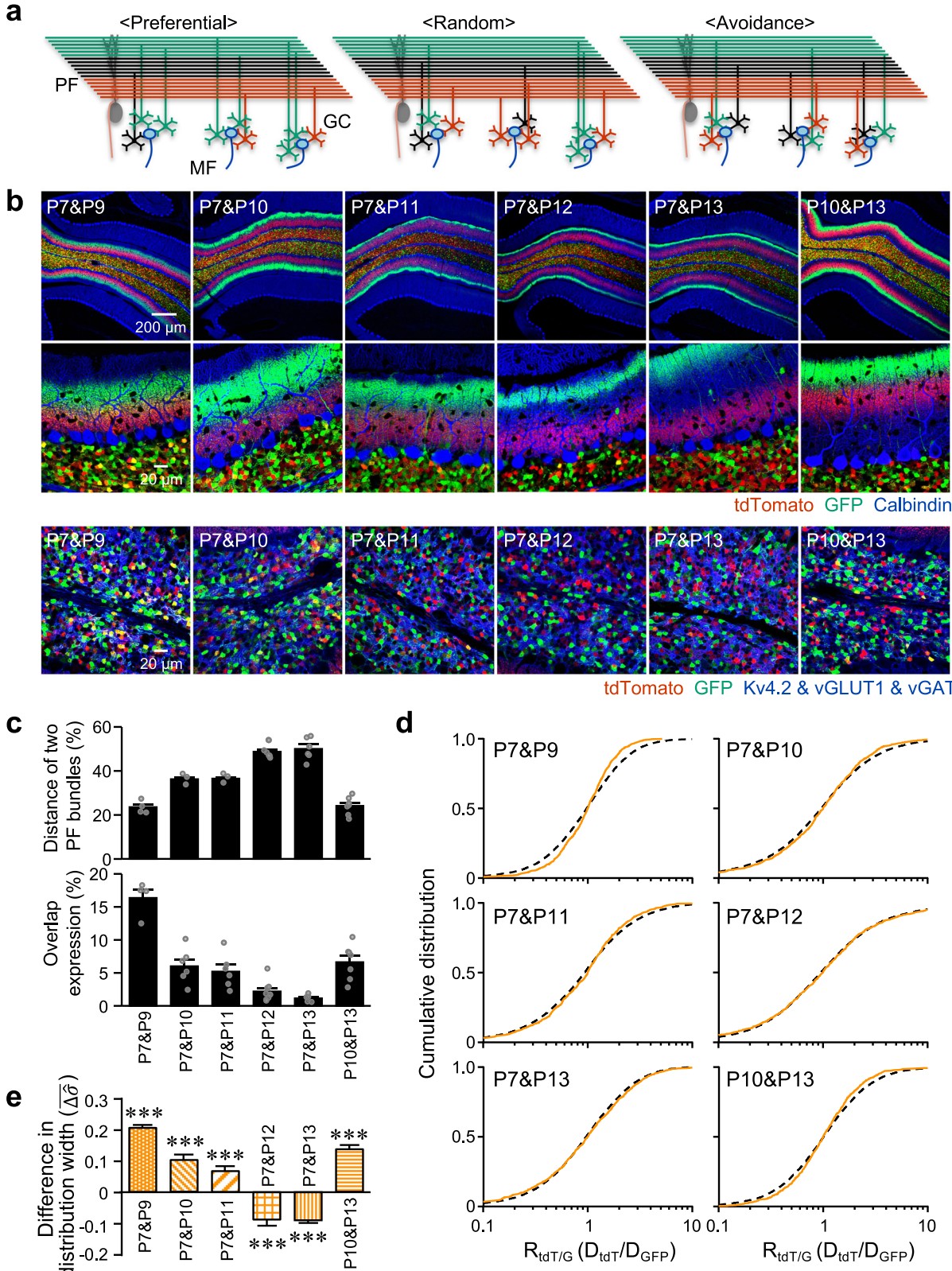

dendrite ID numbers within individual glomeruli (Supplementary Fig. 6; see Methods). The permutation method was used to maintain distributions of the number of GC dendrites connecting to individual glomeruli, as shown in Fig. 4a. As a result of permutation, the average proximity index (PI; see Methods) of ID numbers of the GC dendrites in individual glomeruli was larger in the preferential connection and smaller in the avoidance

connection than the random connection (Fig. 4b), confirming three different patterns of connections. The severer level of preferential or avoidance connections was also made by repeating more permutation trials (lighter magenta or cyan in Fig. 4b). In the experiments, single injection of AAV-GABRα6-GFP resulted in GFP expression in approximately 30% of GCs (Fig. 4c, open bars) and existence of GFP-positive dendrites in 81.0–84.1% of

**Fig. 3 Experimental analyses of overall synaptic connections between MF terminals and GC dendrites. a** Diagrams of three possible connections between MF terminals and groups of GC dendrites. Preferential: glomeruli preferentially include dendrites of GCs with PFs located in a bundle (e.g., S-GCs shown in green). Random: glomeruli randomly include GC dendrites. Avoidance: individual glomeruli tend to avoid including dendrites of GCs with PFs located in a bundle. **b** Confocal images of cerebellar slices expressing tdT (red) and GFP (green) in different groups of GCs. The expression was triggered by the injection of AAV-GABRα6-tdT and AAV-GABRα6-GFP on the indicated postnatal days. Slices were stained (blue) with a calbindin antibody (top and middle panels), or with antibodies of Kv4.2, vGLUT1 and vGAT (bottom panels). **c** Distance of two PF bundles (top), and the percentages of overlap of GCs among all GCs (bottom), when two groups of GCs were labeled at various intervals by double injection ($N = 4$–9 mice, 41 mice in total, 3–5 images per mouse for calculating the percentage of GC overlap). Gray circles represent individual data points. **d** The $R_{tdT/G}$ in the experiments ($D_{tdT}/D_{GFP}$) are plotted as cumulative distributions. Two groups of GCs were labeled by double injection at different times, as indicated in the panels. Orange solid lines show the experimental results ($N = 4$–9 mice, 41 mice in total, 3–5 images per mouse) and dotted black lines show data of shuffled pairs. The x-axis is in log scale. **e** Comparisons of distribution broadness between experimental results and shuffled data. Distribution broadness was described by standard deviations ($\hat{\sigma}$) estimated from lognormal distribution fitting, and comparisons were made by calculating differences in estimated standard deviations ($\bar{\triangle}\hat{\sigma}$). Individual data points are shown in Supplementary Fig. 5b. *$p < 10^{-15}$ (P7&P9), ***$p = 6.35 \times 10^{-5}$ (P7&P10), ***$p = 9.63 \times 10^{-5}$ (P7&P11), ***$p = 1.38 \times 10^{-6}$ (P7&P12), ***$p = 5.18 \times 10^{-10}$ (P7&P13), ***$p < 10^{-15}$ (P10&P13), one-sample Wilcoxon signed rank test for comparison with 0. Data in (**c**) and (**e**) are presented as mean ± s.e.m.

glomeruli (Fig. 4c, orange filled bars), regardless of the time of injection. Consistent with the experimental results, more than 80% of glomeruli in any connections of network models included the dendrites arising from 30% of GCs with consecutive IDs (Fig. 4d) that correspond to a group of GFP-positive GCs after the single injection. Thus, our models were confirmed to cover the basic properties of three types of connections.

To figure out which model most appropriately reproduces the experimental results of $R_{tdT/G}$ distributions, the experimental labeling of different groups of GCs was mimicked in these models. A group of GCs consisted of 30% GCs with consecutive IDs, and different groups of GCs were labeled by tdT and GFP by taking into account the realistic ratios of doubly labeled GCs shown in Fig. 3c. $R_{tdT/G}$ in the model ($R_{tdT/G\text{-model}}$) was calculated from the number of dendrites in individual glomeruli ($N_{tdT}/N_{GFP}$), in which the variabilities in fluorescence intensity observed in experiments were applied (see Methods). We found that the model of a mild preferential connection reproduced the experimental results well; narrower distributions of $R_{tdT/G\text{-model}}$ were observed in cases of shorter distances between two PF bundles and wider distributions of $R_{tdT/G\text{-model}}$ were observed in cases of longer distances between two PF bundles, compared with distributions of $R_{tdT/G\text{-shuffle}}$ (Fig. 4e). Importantly, only the preferential connection model, not the random or avoidance connection model, showed wider distributions similar to those observed in the experimental results of the P7&P12 or P7&P13 injections. The quantification of differences in the distribution broadness ($\bar{\triangle}\hat{\sigma}$) confirmed the reproducibility of the experimental results by the preferential connection model (Fig. 4f), in terms of the negative values observed at P7&P12 and P7&P13 injection. Thus, we conclude that the synaptic connections of GC dendrites with MF terminals were organized by preferential connections. In other words, the synaptic connections are biased in a way that MF terminals preferentially connect with GCs having nearby PFs.

**Properties of synaptic connections between MF terminals and GC dendrites.** The abovementioned analyses of experimental observations and computational models demonstrated mild preferential synaptic connections between MF terminals and GC dendrites. To further characterize the synaptic connections, we performed the deeper analyses by using our computational models. As described above, the experimental results of $R_{tdT/G\text{-data}}$ distributions suggested that the dendrites of D-GCs and S-GCs tend to be segregated in different glomeruli, yet the dendrites of D-GCs and M-GCs or of M-GCs and S-GCs tend to gather together in the same glomeruli. However, due to the inevitable double-labeling of some GCs with tdT and GFP, the $R_{tdT/G\text{-data}}$ values may have moved toward one, resulting in narrower

distributions than the case without the double-labeling. Therefore, we labeled two groups of GCs, D-GCs and M-GCs (near GCs, left in Fig. 5a) or D-GCs and S-GCs (far GCs, right in Fig. 5a), in the model without the double-labeling, and tested $R_{tdT/G\text{-model}}$ distributions. As expected, the $R_{tdT/G\text{-model}}$ distributions in the random connections were indistinguishable between different labeling pairs, near and far GCs (Fig. 5b). The $R_{tdT/G\text{-model}}$ distributions in the preferential and avoidance connections were wider and narrower, respectively, than those in the random connections (Fig. 5c). The extent of differences in the distributions varied according to the level of preferential or avoidance connections: the distribution was slightly different in the mild level of organized connections, but was remarkably different in the severe level of organized connections (Fig. 5c). Nevertheless, the directions of distribution changes, wider in preferential connections and narrower in avoidance connections, were conserved. Notably, the directions of distribution changes were still the same even in the comparison of near GCs (solid lines in Fig. 5c), corresponding to D-GCs and M-GCs, even though it was to a lesser extent than the comparison of far GCs (dotted lines in Fig. 5c). Specifically, such properties were also maintained in the mild preferential connection model (upper left panel in Fig. 5c), which could well reproduce experimental results by considering the double-labeled GCs (Fig. 4e, f). Thus, the analysis using our network models demonstrated that the dendrites of two GCs with shorter distances between PF bundles, such as D-GCs and M-GCs, also tend to be segregated in different glomeruli, although their segregations are gentler than the dendrites of D-GCs and S-GCs. In conclusion, our hypothetical network models could well capture the mild heterogeneity in overall connections between MFs and GCs, which might have been overlooked by conventional statistics.

In the present study, we analyzed GC network structures by taking advantage of AAV-GABRα6, which resulted in the labeling of approximately 30% of GCs having nearby PFs. On the other hand, such labeling of 30% of GCs may not provide sufficient resolution to dissect the microstructures of GC networks. Hence, we conducted two types of high-resolution analyses. We first analyzed spatial arrangement of small numbers of GC somas having nearby PFs by using G, F, and Ripley's K functions. For this purpose, we utilized 5–15% of double-labeled GCs with tdT and GFP after injections at P7&P9, P7&P10, or P10&P13 (Fig. 3c), because their somas were clearly observed. As a result, the random distributions of GC somas were mostly conserved (Supplementary Fig. 7a), reconfirming that there is no clear correlation between the locations of GC somas and their PFs. Second, by labeling a small number of GCs having nearby PFs, we attempted to analyze synaptic connections at a higher-resolution. Unlike the detection of double-labeled GC somas, it is

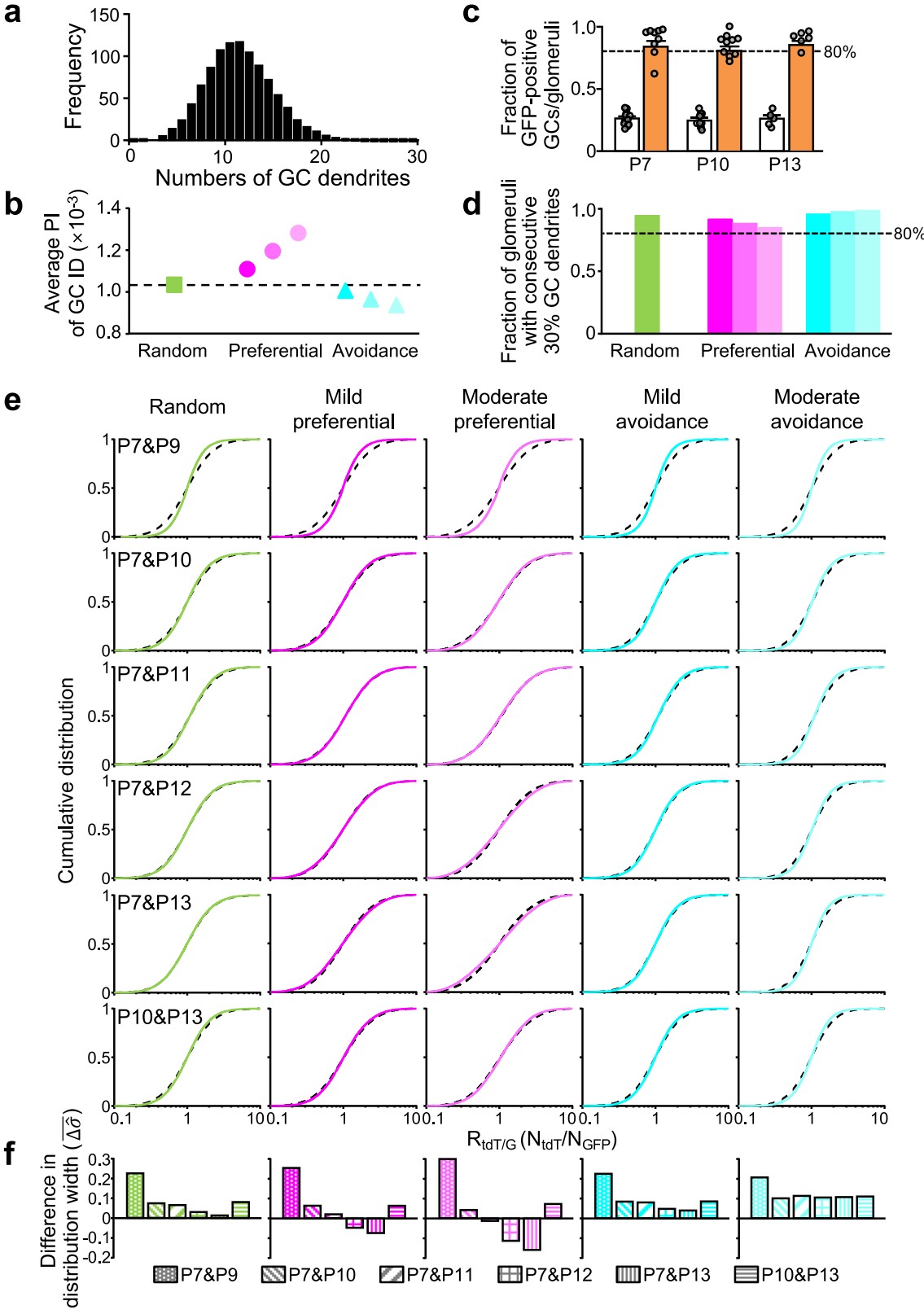

not practical to identify two thin GC dendrites, which could be separately double-labeled after the triple injection, in individual glomeruli. Therefore, we utilized our mild preferential models with labeling of 5% or 15% of GCs having nearby PFs to estimate more details of heterogeneity in the synaptic connections. Except for a comparison between near GCs when 5% of GCs were labeled, the mild preferential model still produced wider

$R_{tdT/G-model}$ distributions than the random network model (Supplementary Fig. 7b–e). These results suggest that biased MF-GC synaptic connections are detectable between GCs having PFs located beyond a distance corresponding to one-fifth to one-fourth (600 to 750 in 3,000 GCs in Supplementary Fig. 7b, e) of all PFs, but not between GCs having PFs located within that distance.

**Fig. 4 Interpretation of overall synaptic connections by network models. a** A histogram of the numbers of GC dendrites included in individual glomeruli of the network model. An averaged histogram of 50 configurations is shown in this figure. **b** The PI of ID numbers of GC dendrites included in individual glomeruli. Averaged values of 50 configurations in each connection model are shown. **c** Experimental data showing percentages of GFP-positive GC somas (open columns) and glomeruli (orange columns) including GFP-positive GC dendrites ($N = 3$–8 mice, 15 mice in total, 3–5 images per mouse). Data are presented as mean ± s.e.m. and gray circles represent individual data points. **d** Fractions of glomeruli including dendrites arising from 30% GCs with consecutive ID numbers. Values are shown as averages of 50 configurations in each connection model. In (**b**) and (**d**), results in severer level of preferential or avoidance model are shown in lighter colors. **e** Cumulative distributions of the $R_{tdT/G}$ ($= N_{tdT}/N_{GFP}$) in the random (green), mild preferential (magenta), moderate preferential (light magenta), mild avoidance (cyan), and moderate avoidance (light cyan) connection models. The x-axis is in log scale. Two groups of GCs are selected so as to be equivalent to the experimental labeling by different times of double injection as indicated in the left panels, in terms of overlaps. Color solid lines show the model results and dotted black lines show data of shuffled pairs. **f** Comparisons of distribution broadness ($\bar{\triangle}\hat{\sigma}$) between model results and shuffled data.

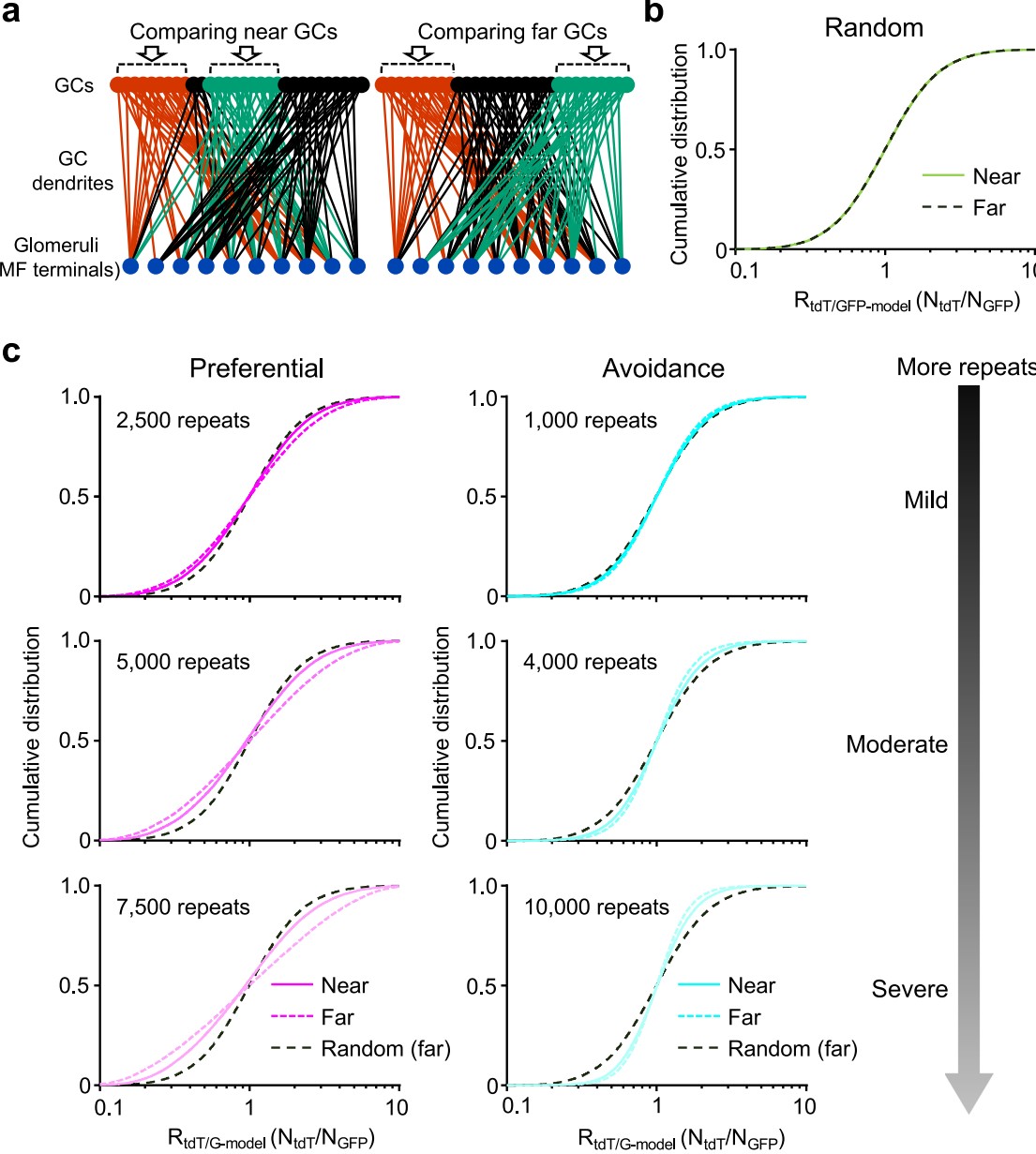

**Fig. 5 Properties of synaptic connections in the network model. a** Diagrams showing the location of two groups of GCs located at near (left, near GCs) or far (right, far GCs) distance in the network model. To see how the distance of two groups of GCs affects the $R_{tdT/G}$ distributions (**b**, **c**), $R_{tdT/G\text{-model}}$ values were calculated by assuming that either far GCs or near GCs were labeled. **b** The overlapping cumulative distributions of $R_{tdT/G\text{-model}}$ calculated using near (solid green line) and far (dotted dark brown line) GCs in the random connection model. **c** The cumulative distributions of $R_{tdT/G\text{-model}}$ in the different levels of preferential (left) or avoidance (right) connection models. When the permutation trials were repeated more, as indicated in the panels, the levels of preferential or avoidance connection became severer. $R_{tdT/G\text{-model}}$ calculated using near (solid lines) and far (dotted lines) GCs are overlaid together with the result in random connection model (dotted dark brown lines). The x-axis is in log scale.

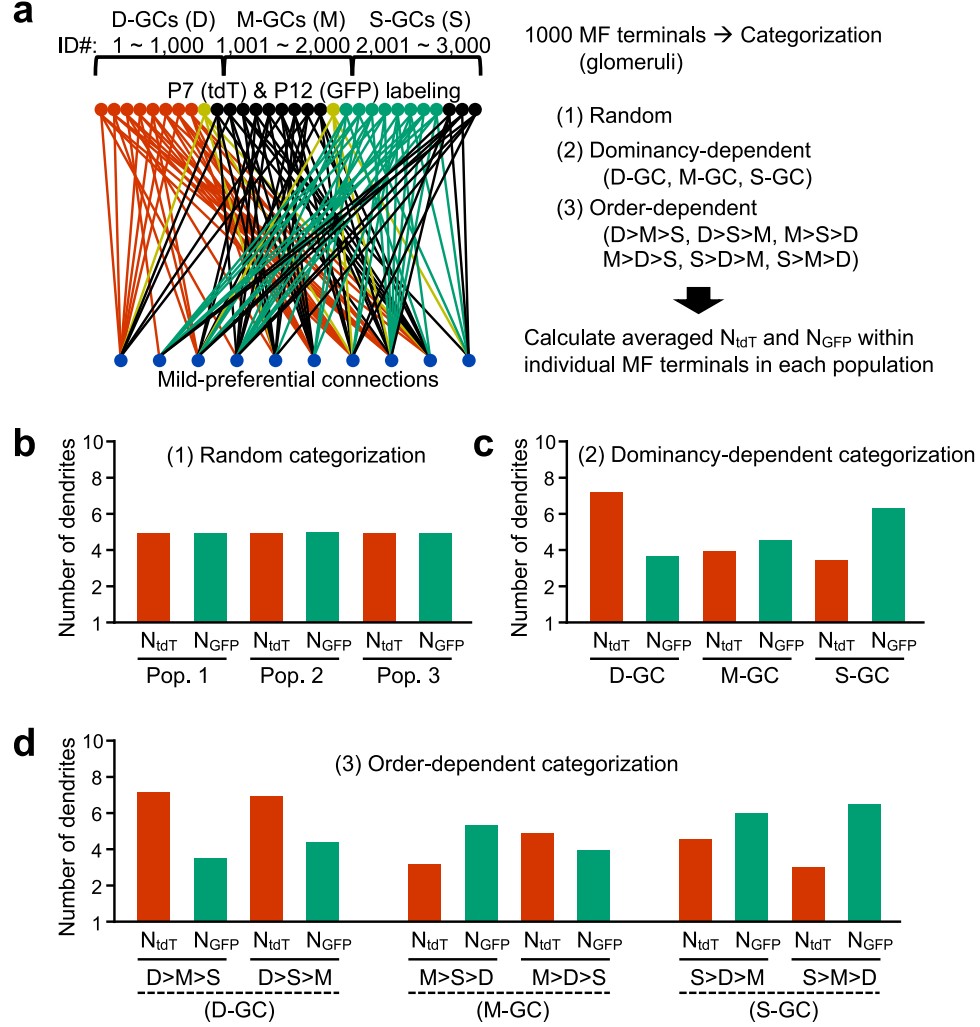

**Fig. 6 Interpretation of synaptic connections with specific MF terminals by mild preferential model. a** Diagram of mild preferential connection model with mimicked labeling of GCs by P7&P12 injection (left). Red, green, and yellow GCs represent GCs expressing tdT, GFP, and both, respectively. GCs are divided into D-GCs, M-GCs, and S-GCs, based on ID numbers. MFs are categorized into populations based on one of three criteria (right), which are made according to groups of GCs connecting. Numbers of tdT ($N_{tdT}$)- and GFP ($N_{GFP}$)-positive GC dendrites within individual MF terminals are then averaged in each population. **b–d** Averaged $N_{tdT}$ and $N_{GFP}$ in different populations of MF terminals, when MF terminals were categorized by random categorization (**b**), dominancy-dependent categorization (**c**), or order-dependent categorization (**d**).

**Synaptic connections with MF terminals originating from specific nuclei: revisited.** Our analyses using network models so far were based on the unbiased sampling of MF terminals, and demonstrated that the MF-GC network structure could be revised beyond uniform random connections. We then utilized our network model of mild preferential connections to understand the synaptic connections in specific samples of MF terminals, with consideration of the experimental results of GC dendrites connecting with DCoN-MFs and PN-MFs shown in Fig. 2. For this analysis, MF terminals in the model were categorized according to one of three criteria, namely, random, dominancy-dependent, or order-dependent categorization, which were defined by the type of GC dendrites connecting with the MF terminals (Fig. 6a and Supplementary Fig. 8). For the random categorization, MF terminals were randomly divided into three populations. For the dominancy-dependent categorization, MF terminals were divided based on dominantly connecting GCs, D-GCs, M-GCs, or S-GCs, resulting in three populations of MF terminals. In the model, GCs with ID numbers of 1–1,000, 1,001–2,000, and 2,001–3,000 were simply regarded as D-GCs, M-GCs, and S-GCs, respectively (Fig. 6a). For the order-dependent categorization, the most and

second most abundant types of GC dendrites connecting with the MF terminals were considered, and consequently the MF terminals were divided into six populations. The MF terminal populations categorized by the order-dependent manner were named based on the descending order of dominance of GC dendrites, e.g., D > M > S stands for a population of MF terminals connecting to D-GC, M-GC, and S-GC dendrites in the descending order. We then counted the numbers of tdT ($N_{tdT}$)- or GFP ($N_{GFP}$)-positive GC dendrites connecting with individual MF terminals and averaged them for each population, under the assumption that GCs were labeled by P7&P12 injection (Fig. 6a), as we experimentally performed together with labeling of PN-MFs and DCoN-MFs in Fig. 2. The analysis demonstrated that both numbers, $N_{tdT}$ and $N_{GFP}$, in any populations of randomly divided MF terminals were equivalent (Fig. 6b). In contrast, $N_{tdT}$ and $N_{GFP}$ varied when MF terminals were categorized in a dominancy-dependent or order-dependent manner (Fig. 6c, d). These results confirm that the varied intensities of tdT and GFP in specific MF terminals presented in Fig. 2 are an indication of structured synaptic connections between the specific MF terminals and GC dendrites according to the PF locations.

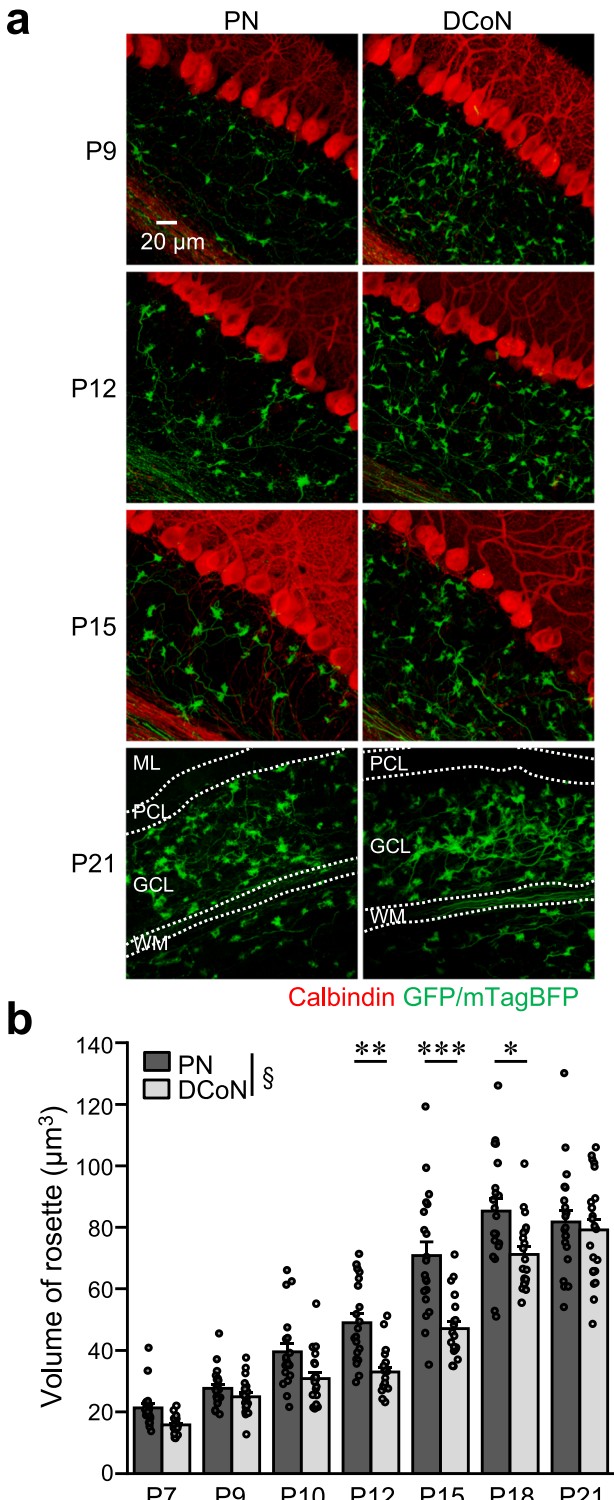

**Fig. 7 Different time courses of increases in MF terminal size between PN-MFs and DCoN-MFs. a** 3D projection images of sagittal cerebellar slices observed at the indicated postnatal days, in which PN-MFs or DCoN-MFs were labeled with GFP (P9, P12, and P15) or mTagBFP (P21). Slices for P9, P12, and P15 were stained (red) with a calbindin antibody. PCL, PC layer; WM, white matter. **b** Volume of terminals (rosette) of PN-MFs (dark gray) and DCoN-MFs (light gray). For comparison of overall time course; $^{§}p = 9.8 \times 10^{-12}$, two-way ANOVA, For comparison between PN and DCoN at each time; $^{**}p = 0.0051$ (P12), $^{***}p = 3.8 \times 10^{-7}$ (P15), $^{*}p = 0.033$ (P18), two-way ANOVA followed by the Bonferroni test ($N = 4$ or 7 mice, 62 mice in total, 3–5 images per mouse). Data are presented as mean ± s.e.m. and gray circles represent individual data points.

Furthermore, to interpret the synaptic connections in PN-MFs and DCoN-MFs, we compared the details of $D_{tdT}$ and $D_{GFP}$ (Fig. 2e) with $N_{tdT}$ and $N_{GFP}$ (Fig. 6c, d). Considering that $D_{GFP}$ was higher than $D_{tdT}$ in both PN-MFs and DCoN-MFs, and that $D_{GFP}$ in DCoN-MFs was higher than $D_{GFP}$ in PN-MFs (Fig. 2e), PN-MFs have similarity with M-GC-dominant MFs and DCoN-MFs have similarity with S-GC-dominant MFs (Fig. 6c). Thus, PN-MFs and DCoN-MFs are considered to predominantly form synapses with M-GCs and S-GCs, respectively. In addition, because $D_{tdT}$ in DCoN-MFs was also higher than $D_{tdT}$ in PN-MFs (Fig. 2e), PN-MFs likely included more M > S > D types than M > D > S types among M-GC-dominant MFs, and DCoN-MFs likely included more S > D > M types than S > M > D types among S-GC-dominant MFs (Fig. 6d). Therefore, it is speculated that the MF inputs from different nuclei are mainly conveyed to different sublayers of MLs, at least in part owing to the synaptic organization from specific MFs to GCs in the GCL.

**Contributions of the developmental process to organize the biased synaptic connections between MF terminals and GC dendrites.** In the present study, we found that overall synaptic connections from MFs to GC dendrites are biased in a way that GCs having nearby PFs tend to share the same MF inputs, and specific MF terminals originating from PN and DCoN most preferentially connect to GCs having PFs located at different sublayers. However, it remains to be elucidated as to how the organized synaptic connections are formed. Considering that GCs are gradually made in order from D-GCs to S-GCs during postnatal development[18], the organized synaptic connections might in part rely on enhanced synaptic formation between developmental timing-matched MF terminals and GC dendrites. This raises a possibility of asynchrony in the developmental time courses of MFs depending on their origins. Specifically, PN-MFs preferentially connecting with M-GCs may be developed earlier than DCoN-MFs preferentially connecting with S-GCs. We tested this possibility by analyzing the volume of MF terminals during postnatal developmental periods, because it has been demonstrated that MF terminals increase in size and surface area during the developmental periods[23]. MFs originating from the PN or the DCoN were labeled by injecting AAV-sSyn-GFP or -mTagBFP into these nuclei, and their typical morphologies of terminals, called rosettes, were visualized using 3D imaging (Fig. 7a). Consistent with the previous study[23], quantification of these MF terminals demonstrated that their volumes gradually increased during postnatal development (Fig. 7b). Interestingly, the time courses of the increase in the volume of MF terminals were different between the PN-MFs and the DCoN-MFs, and consequently PN-MF terminals were significantly larger than DCoN-MF terminals during a period from P12 to P18, but not before P10 and at P21 (Fig. 7b). Thus, the development of PN-MF terminals appears to precede the development of DCoN-MF terminals. This sequence fits with the concept of enhanced synaptic formation between developmental timing-matched MF terminals and GC dendrites, given the earlier maturation of M-GCs than S-GCs.

## Discussion

We determined aspects of the structural association between GC networks in the GCL and the ML in the cerebellum using our labeling technique. Observations of labeled GC soma distributions confirmed that they were not clustered, but were randomly distributed, in the GCL of both sagittal and transverse slices, even though their PFs were bundled in the ML. Despite the random distribution of GC somas, analyses of acquired images as well as the computational network model identified mild preferential overall synaptic connections between MF terminals and GC

dendrites, indicating that GCs having PFs located close to each other tend to receive inputs from the same MF terminals. Our analysis of specific MF terminals showed that PN-MFs and DCoN-MFs preferentially formed synapses with M-GCs and S-GCs, respectively, among the three types of GCs, implying that the overall preferential connections may be a consequence of MF origin-dependent synaptic connections, and vice versa. Furthermore, we found that an increase in the size of PN-MF terminals preceded that in the size of DCoN-MF terminals during postnatal development, suggesting earlier development of PN-MFs than that of DCoN-MFs. Given that M-GCs are made before S-GCs, it appears to be reasonable that synaptic connections principally from PN-MFs to M-GCs and from DCoN-MFs to S-GCs are formed during the developmental period. Thus, our study demonstrated biased synaptic connections from MFs to GCs according to MF origins and PF locations, and presented a potential mechanism that developmental process may contribute to the formation of such synaptic connections.

A previous study using mosaic analysis demonstrated that PFs from GC clones derived from a single progenitor cell were restricted to a specific sublayer of the ML, whereas their GC somas appeared random across the anterior-posterior axis in the GCL[9]. GCs labeled by electroporation at different postnatal days had their PFs stacked together at specific sublayers of the ML, but their somas were distributed throughout the GCL[11]. Another study found that neighboring GC somas in the GCL labeled with calcium indicator gave rise to sparsely distributed PFs[10]. Our previous results obtained by using the AAV-GABRα6 were consistent with these studies, in terms that there is no association in the anterior-posterior axis between positions of GC somas in the GCL and PF locations in the ML[12]. In addition, our present study demonstrated random distributions of GC somas also in the dorsal-ventral or lateral axis, regardless of their bundled PFs. After exiting from the proliferation in the external granular layer (EGL) during postnatal development, GCs migrate tangentially within the inner EGL and then radially along Bergmann glial fibers[24,25]. Given that each GC's position in the dorsal-ventral or lateral axis becomes fixed when the radial migration begins, it may be reasonable that GCs migrating at the same time are dispersed to avoid competing for their paths of radial migration. On the other hand, the anterior-posterior distributions of GC somas appear to be determined by the positions where GCs stop migrating. A study using fresh cerebellar slices obtained at P10 showed that the majority of GCs reached the deep GCL through rapid, radial, and unidirectional migration within the GCL[26], suggesting that GCs may stop migrating in response to some molecular cues enriched at the deep GCL. However, dispersed distributions of GCs migrating at the same time raise the possibility that GCs may respond to the cues in wider areas in vivo. Another study discovered molecules in GCs involved in the cessation of migration: the SnoN1-FOXO1 transcriptional complex regulates the cessation through the repression of doublecortin that has been considered to be critical for neuronal migration and morphology[27]. It would be interesting in the future to understand the identity of cues and their association with the transcriptional complex.

Considering the short dendrites of GCs, the random GC soma distributions are superficially appropriate for the random synaptic connections with presynaptic MF terminals. Nevertheless, our analyses showed preferential synaptic connections, raising the hypothesis that there may be some active mechanisms to make specific synaptic connections, and such active mechanisms together with the geometrically passive consequences of the random GC soma distributions may result in the mild preferential connections. Because PN-MFs and DCoN-MFs preferentially form synapses with M-GCs and S-GCs, respectively, their active mechanisms are thought to be associated with the different properties of MFs according to their origins and of GCs according to the locations of their PFs. A easily noticeable difference among GCs is their birth timing, i.e., early-born GCs have PFs in the deeper layers and later-born GCs have PFs in the more superficial layers[18]. It has been reported that in several brain regions, clonally-related neurons or neurons sharing developmental time windows form specific networks[28–31]. However, whereas GCs are gradually produced during the first two postnatal weeks, neurons projecting MFs are made at the embryonic stage, and MFs arrive in the cerebellum by P0 and go through dynamic processes before making synapses with GCs[32,33]. Instead of their birth date, temporally matched synaptogenesis between a specific origin of MFs and a specific group of GCs having PFs in a certain sublayer may underlie the formation of organized synaptic connections. In the present study, we indeed observed an earlier developmental process in PF-MFs than DCoN-MFs. Although it remains unclear whether the timing of synaptogenesis is truly well matched between PF-MFs and M-GCs or between DCoN-MFs and S-GCs, this order of sequential development of MFs is consistent with the order of GC development. Therefore, preferential synaptic connections appear to be partly attributable to the gradual developmental process of MFs and GCs.

While we utilized the strength of AAV-GABRα6 that enables PF location-dependent labeling of GCs, its limitations have to be recognized. One such limitation might be the inadequate resolution to analyze the detailed structures of GC networks due to the labeling of approximately 30% of GCs. By taking advantage of tdT/GFP double-labeled GCs, we reconfirmed the random distributions of GC somas, despite their bundled PFs. Although our analysis in the mild preferential connection model with a small number of labeled GCs predicted the detailed network structures, it would be ideal to experimentally confirm such structures by developing a high-resolution GC labeling technique. Despite the limitation, we could find the differences in GC connectivity with MFs according to their PF locations. In addition to the connectivity, other properties in GCs may also be organized according to their PF locations. Evident heterogeneity in overall GCs has been reported regarding molecular expression[34], morphology[35] and electrophysiological properties[8,36–38]. We previously succeeded to combine calcium imaging using indicator dyes with AAV-GABRα6-mediated labeling of GCs, and found non-uniform calcium responses in GCs associated with the location of their PFs[12]. Combining the analyses to detect the heterogeneous properties with our labeling technique using AAV-GABRα6 would have the potential to correlate these properties of GCs with their PF locations.

Based on early studies using Golgi staining[5,6], it was hypothesized that signals arising from MFs of different origins may be conveyed differently to the ML through sublayers of activated PFs[7]. This hypothesis has been challenged by the studies demonstrating no associations between GC soma locations and their PF locations[9–12]. A recent study directly tested associations between synaptic organizations from MFs to GCs in the GCL and sublayers of PF projections in the ML[13]. This study using lobule VI demonstrated quantitative differences in synaptic connections of early- and late-born GCs, corresponding to D-GCs and S-GCs, respectively, with MFs originating from specific nuclei. Consistent with this study, we also found non-uniform MF-GC synaptic connections in lobule IV/V by analyzing amounts of tdT-positive D-GC dendrites and GFP-positive S-GC dendrites in PN-MFs or DCoN-MFs. However, detailed properties of synaptic connections were different between these two studies. The previous study demonstrated that more neurons in basal PN provided inputs onto S-GCs than D-GCs, while more neurons in some motor and sensory nuclei, including a part of DCoN, provided inputs onto D-GCs than S-GCs. In contrast, our study suggested

that PN-MF terminals preferentially connect with dendrites of M-GCs, while DCoN-MF terminals preferentially connect with dendrites of S-GCs. Two explanations are possibly considered to reconcile the discrepancy. First, different lobules may present different properties of synaptic connections, since generations of GCs appear to be differentially regulated in each zone of vermis or hemisphere[39,40]. A previous study reported that GC generations in the central zone, including lobule VI, were delayed relative to other zones[40]. Combining this report with the implication of our results, the potential contribution of developmental process to the synaptic connectivity organization, the discrepancy in synaptic connectivity may result from the different developmental time courses of GCs in lobule IV/V and VI. Second, the differences may arise due to different means of analyses. The previous study used a measurement of labeled cells in precerebellar nuclei by monosynaptic retrograde rabies virus tracing, whereas we observed labeled GC dendrites in MF terminals. Although further studies are required to understand the organization of synaptic connections between MFs and GCs in the entire cerebellum, these two studies provided direct evidence that MF-GC synaptic connections are at least in part organized according to sublayers of PF projections.

The structural organization of MF-GC synaptic connectivity implies two features of information coding through GC networks. One is that a certain type of information can be conveyed mainly to a cluster of PF activities. A previous study indeed demonstrated that the clustered PF activities were triggered upon a type of sensory stimuli[10]. This study also demonstrated that non-correlated PFs, which were likely driven by different MFs, were also clustered in space, suggesting that different MFs carrying the same sensory information innervate GCs having neighboring PFs. Because different neurons in a certain precerebellar nucleus could have adjacent MFs in the same lobule of the cerebellum[41,42], MF terminals from the specific nuclei in our analysis likely arose from not only collaterals of single neurons but also different neurons. We then found that these MF terminals preferentially, but not specifically, connected to GCs having neighboring PFs. Although further analyses are necessary to precisely distinguish the preferential connections with collateral MFs or different MFs, our study supports the structural organization suggested by the study showing clustered PF activities[10].

The second feature is that different types of information can be separated according to the locations of clustered PF activities. The cerebellar network in the GCL has been thought to be suitable for pattern separation[43,44]. Specifically, a very large number of GCs and sparse synaptic connectivity from MFs to GCs have been theoretically shown to be beneficial to demix input signals entering through MFs. However, recent studies using in vivo calcium imaging have challenged this idea by showing higher densities of GC activations than predicted[45–47]. Our finding of preferential connections, in which MFs originating from a certain precerebellar nucleus preferentially innervate GCs having neighboring PFs, implies higher chances for the simultaneous activation of two or more dendrites of individual GCs when a specific type of input signal enters. This is expected to result in higher densities of GC activations, compared with the traditionally accepted assumption of random synaptic connectivity. Additionally, the preferential connections are thought to distinguish different types of information by the locations of the clustered PF activities. It has not yet been identified as to whether different locations of clustered PF activities would cause any differences in Purkinje cell activities. However, the clustered PF inputs are considered to be efficient for triggering dendritic spikes[48], dendritic calcium increase[49,50], endocannabinoid release[51], or synaptic plasticity[52,53], owing to the synaptic integration of PF inputs in the Purkinje cell dendrites. We hence propose that preferential synaptic connections create conditions in which these events are easily triggered in different regions of Purkinje cell dendrites, and cerebellar networks utilize this system to separate different contextual information.

## Methods

**Surgery for the stereotaxic injection of AAV into mouse brain**. All procedures involving mice were performed according to the guidelines of the Institutional Animal Care and Use Committee of Korea Institute of Science and Technology (Approval number: KIST-5088-2022-03-041). Because of the nursing ability, ICR mice of both sexes were used in this study. Time course of experiments from AAV injection to fixation for the imaging analyses was summarized in Supplementary Fig. 1a. AAV serotype 1 vectors, with estimated titers of approximately $10^{13}$ vector genome copies, were produced as described previously[14]. To trigger the expression in GCs, AAV vectors with GABRα6 promoter (about 1.5 μl total) were stereotaxically injected into cerebellar lobule IV/V of 7 to 13-day-old or 22-day-old mice, by the procedures previously described[14]. For the second injection, the hole made on the cranium at prior injection was reused. To trigger the expression in the DCoN- or PN-MFs, AAV-sSyn (about 200 nl total) was injected into these nuclei of 1 to 2-day-old mice for the analysis before P18 or of 12-day-old mice for the analysis at P21. Coordinates for the DCoN were about 2.0 or 2.5 mm caudal from lambda, 0.8 or 1 mm lateral from the midline and 2.1 or 2.7 mm deep from dura, and coordinates for the PN were 0.15 or 0.2 mm caudal from lambda, 0.28 or 0.35 mm lateral from the midline and 3.7 or 4.9 mm deep from dura in 1 to 2-day-old or 12-day-old mice, respectively. After the surgery, mice were kept on a heating pad until they recovered from the anesthesia and then they were returned to their home cages. In case of labeling mitotic cells, EdU (100 mg/kg body weight, Thermo Fisher Scientific) was intraperitoneally injected.

**Patch clamp recording**. Chemicals used were obtained from Sigma or Wako Pure Chemical Industries, unless otherwise specified. Slice preparation and whole-cell patch clamp recording from PCs were performed as previously described[15]. Sagittal cerebellar slices (200 μm) were obtained from P21 to P25 mice subjected to the double injection of AAV-GABRα6-tdT at P7 and AAV-GABRα6-GFP at P12, and stored in extracellular solution (ACSF) containing the following (in mM): 125 NaCl, 2.5 KCl, 1.3 MgCl₂, 2 CaCl₂, 1.25 NaH₂PO₄, 26 NaHCO₃, and 20 glucose. During the recording, 0.01 mM bicuculline methochloride (Tocris Bioscience, 0131) was added to the ACSF. For whole-cell recordings from PCs, electrodes (5-10 MΩ) were filled with the internal solution containing the following (in mM): 130 potassium gluconate, 2 NaCl, 4 MgCl₂, 4 Na₂-ATP, 0.4 Na-GTP, 20 HEPES [pH 7.2], and 0.25 EGTA. To visualize recording PCs, Alexa633 was included in the internal solution. EPSCs were evoked in PCs (holding potential of –70 mV) by activating PFs with a glass stimulating electrode on the surface of the ML (PF-EPSCs). To identify areas of PFs expressing tdT or GFP, their fluorescence was identified under a microscope (Olympus BX61WI or Nikon FN1). The stimulating electrode was placed either on tdT- or GFP-positive areas, to stimulate P7-PFs or P12-PFs, respectively. PF-EPSCs were acquired and analyzed using pClamp software (Molecular Devices) and custom Python script with pyABF package (Harden, SW (2022). pyABF 2.3.5. [Online]. Available: https://pypi.org/project/pyabf). Rise time and decay time were estimated from time constant obtained by a single exponential fitting to PF-EPSC traces before and after the peak, respectively.

**Immunohistochemistry, imaging, and western blot**. Primary antibodies used were mouse anti-calbindin (Sigma, C9848), mouse anti-NeuN, (Millipore, MAB377), rabbit anti-vGLUT1 (Synaptic Systems, 135303), rabbit anti-vGAT (Synaptic Systems, 131003), mouse anti-Kv4.2 (Antibodies Inc., 75–016), rabbit anti-GABRα6 (Synaptic Systems, 224603), and mouse anti-β-Actin (Santa Cruz Biotechnology, sc-47778). Secondary antibodies used were Alexa Fluor 647-conjugated anti-mouse IgG (Life Technologies, A21236), Alexa Fluor 405 or 488-conjugated anti-rabbit IgG (Life Technologies, A31556 or A11034), or HRP-conjugated anti-mouse or anti-rabbit IgG (GE Healthcare, NA931V or NA9340V).

Mice subjected to AAV injection were anesthetized at P21–P23 (P7–P13 AAV injection) or at P32 (P22 AAV injection), and perfused transcardially with 4% paraformaldehyde (PFA) in 0.1 M sodium phosphate buffer (pH 7.4). For analyses of MF terminal volume, mice at designated postnatal days were anesthetized and perfused transcardially with 4% PFA (P12, P15, and P18) or only with PBS (P7, P9, and P10). To see the expression of endogenous GABRα6, mice without AAV injection were perfused transcardially with 4% PFA (P11 and adult) or only with PBS (P6). The cerebellum of all ages of mice was immediately removed and then post-fixed with 2% PFA, followed by making sagittal or transverse slices (40 or 50 μm) using a vibrating microtome (Leica VT 1200 S). Immunohistochemistry was performed by following a protocol described previously[14]. EdU was detected using the Click-iT EdU Alexa Fluor 647 Imaging Kit (Molecular Probe, C10640). Single optical sections or z-stacks of confocal images were acquired by an A1R laser scanning confocal microscope (Nikon). Images were taken in the three-fourths of external side of lobule IV/V at center of vermis, where the expression of fluorescent molecules was nearly evenly observed. The mice with off-target AAV injection, which were clearly visualized by the location of fluorescent molecule expression, were excluded from the analyses.

For western blot analysis, the cerebellum of all ages of mice was dissected out, and homogenized in the lysis buffer (50 mM Tris-HCl [pH 8.0], 150 mM NaCl, 1 mM EDTA, 0.2% NP-40 Alternative (Calbiochem), and 10% glycerol). After removal of the nuclear fraction by centrifugation at 15,000 rpm for 30 min, the supernatants were used as protein samples, and subjected to SDS-PAGE and western blot analysis to detect endogenous GABRα6 and β-Actin expression using chemiluminescent detection reagent (GE Healthcare, RPN2232). Protein samples loaded were 7.5 µg/lane. The chemiluminescence signals were captured using ImageQuant LAS 4000 (GE Healthcare). Two samples were separately prepared from two mice for each age group. Uncropped and unedited blot images of Supplementary Fig. 2e are presented in Supplementary Fig. 9.

**Image analysis software**. Image analyses were performed using NIS Elements (Nikon), ImageJ (National Institutes of Health), or custom MATLAB (Math-Works) script.

**Quantification of the location of labeled PF bundles**. Numbers of mice used for these analyses were 117 in total and 4–17 for each time of single injection, or 4–9 for each pair of double injection. Line scan profiles (width of 248 µm) were obtained across the layers in low magnified images ($1.27 \times 1.27$ mm$^2$). Line scan profiles from five different areas were taken from one mouse, and their average is used as one example. The profiles of GFP or tdT in the ML were fitted with the Gaussian function, and the position of the peak was considered as a center of the labeled PF bundles. The distance of the labeled PF bundles from PC soma or of two PF bundles was shown as percentages in the whole length of PC soma and dendrites, which were measured in the calbindin images.

**GC expression percentage and distributions of labeled GC somas**. The threshold of GFP, tdT, or EdU signals in GCL images was determined by the automatic thresholding of ImageJ software (RenyiEntropy). All GC somas, including labeled and unlabeled GC somas, were detected by the spot detection functions of NIS Elements in the images of GCLs stained with a Kv4.2 antibody, which was used to visualize GC membranes. In case if more than 80% areas in detected spots were labeled with fluorescent molecules over the threshold, the spots were considered as labeled GC somas. The percentages of GFP- or tdT-positive GC somas among all GCs or EdU-positive GCs were calculated to estimate the expression percentages. The positional information of GFP-labeled or tdT/GFP double-labeled GC somas was used for the analyses in G, F, and Ripley's K functions.

**Generation of random and clustered distributions of GC somas**. Spatial distribution of GFP-positive GC somas is not a typical spatial point process on 2-dimensional free space, but rather it is more plausible to be confined on the irregular lattice formed by whole population of GC somas. We, therefore, generated the random or clustered distributions of a group of GC somas on the irregular lattice, which were made by reconstructing the location of all GC somas detected in confocal images (Supplementary Fig. 3a-i, ii). To generate random configurations, an equivalent number of GCs as GFP-positive GCs was computationally selected (sGCs) by Bernoulli trial for each GC (Supplementary Fig. 3a-iii). Such selection was repeated 50 times in a single image, so that 50 examples of random configurations were obtained from a single image. To generate clustered configurations from individual random configurations, a pair of sGCs and non-sGCs was randomly picked in a random configuration. The location of these GCs was permutated under the following criteria: ratios of other sGC numbers to other non-sGC numbers were calculated within circles (a radius of 8 µm) centered at the picked GCs, and permutation was executed if the ratio around the picked sGC was smaller than the ratio around the picked non-sGC (Supplementary Fig. 3a-iv). A single clustered configuration was made by repeating the permutation trials, from the pair picking to the decision of execution, 20,000 times (Supplementary Fig. 3a-v), which bring the permutation events to a plateau level (Supplementary Fig. 3b).

**Analysis of spatial randomness in GC soma distribution by the G, F, and Ripley's K functions**. The estimations of G, F, and Ripley's K functions, denoted as $\hat{G}$, $\hat{F}$, and $\hat{K}$ functions, indicating that those functions are the estimations of distributions from sample statistics, were used to analyze the distribution of GFP-positive GC somas (Fig. 1d, e), or tdT/GFP double-positive GC somas (Supplementary Fig. 7a), in the original images as well as computationally generated random or clustered configurations. The boundary of the areas in individual images used for the analysis was assumed by convex hull of spatially distributed all GCs. The nearest neighbor methods, G and F functions, were estimated by calculating cumulative distribution of minimum event-to-event distances and minimum fixed-point-to-event distances, respectively (Supplementary Fig. 4a, b)[54]. The term "event" indicates our target of analysis, GFP-positive GC somas, and "points" indicate randomly distributed arbitrary points generated by binominal process within the boundary. The nearest neighbor distances were collected from all images of the same injection times to build estimated $\hat{G}$ ($\hat{G}(d)$) and $\hat{F}$ ($\hat{F}(d)$) functions. The $\hat{G}(d)$ and $\hat{F}(d)$ values ($\hat{G}_{data}(d)$, $\hat{G}_{rand}(d)$, or $\hat{G}_{cls}(d)$, and $\hat{F}_{data}(d)$ $\hat{F}_{rand}(d)$, or $\hat{F}_{cls}(d)$) were interpolated and re-plotted against the values in random configurations

($\hat{G}_{rand}(d)$ and $\hat{F}_{rand}(d)$, Supplementary Fig. 4c), to compare all data from different times of injection together. To avoid the locality problem in nearest neighbor methods, we also used Ripley's K function[55], which was estimated by counting numbers of events within a given radius ($r$) of circles. The estimation of function could be formulated as follows:

$$\hat{K}(r) = \frac{1}{\hat{\lambda}} \cdot \frac{1}{N} \sum_{i=1}^{N} \sum_{j \ne i} I\left(r_{ij} < r\right) \qquad (1)$$

where $N$ is the number of events, $\hat{\lambda}$ denotes the estimated event density calculated by dividing $N$ by the area within boundary, and $r_{ij}$ denotes event-to-event distance. $I(x)$ is the function that gives 1 if $x$ is true, and 0 if $x$ is false. Averages of the $\hat{K}(r)$ obtained from all images of the same injection times are presented in figures.

Numbers of mice used for the analyses were 50 in total and 3–5 for each time of injection. Three different images in GCL ($211.7 \times 211.7$ µm$^2$) per one mouse were taken and used for the analyses. Analyzed results in computationally created random and clustered configurations are overlaid in figures (Fig. 1d, e and Supplementary Fig. 7a), yet only single line of results is shown for random configurations, except for $\hat{K}$ functions in Supplementary Fig. 7a, because all other results for random configurations are the same regardless the time of injection.

**Semi-automatic segmentation of individual glomeruli**. We built a semi-automatic segmentation script in the MATLAB performing the following procedure and used it for the segmentation of individual glomeruli. The segmentation was performed using two kinds of GCL images, one stained with a Kv4.2 antibody (GC dendrites) and another stained with antibodies of vGLUT1 (MF terminal) and vGAT (Golgi cell axons) (Supplementary Fig. 5a). Wiener filter was used for denoising, and Contrast Limited Adaptive Histogram Equalization (CLAHE)[56] was applied to the images to correct the local brightness and contrast differences (Supplementary Fig. 5a-i). The procedures in semi-automatic segmentation program were as follows: (1) The images of Kv4.2 and vGLUT1/vGAT were summed (Supplementary Fig. 5a-ii), and glomerulus cluster areas were detected by thresholding the summed images (Supplementary Fig. 5a-iii). (2) Because the centers of individual glomeruli could be observed as voids in the glomerulus cluster areas of Kv4.2 images, such centers were detected by H-minima transformation[57] and size filtering (Supplementary Fig. 5a-iv). (3) The detected centers were used as markers to filter the segmented regions by watershed algorithm[58] (Supplementary Fig. 5a-v). The threshold values and other parameters were manually determined, to minimize the erroneous segmentation. In case if the semi-automatically segmented regions were made clearly by errors, the regions were manually removed from the analyses. We did not manually add missing segmentations to reduce the chances of biased sampling. Numbers of mice used for the analyses were 91 in total and 4–11 for each time of injection. Three to five different areas of images ($211.7 \times 211.7$ µm$^2$) in individual mice were used for the segmentation. Total glomeruli used for the analyses of GC dendrites were 772–1121 for each time of injection.

**Analysis of the synaptic connections of GC dendrites in the individual glomeruli**. Fluorescent intensities of GFP or tdT were measured in the individual glomeruli, which were detected by the segmentation program described above.

In the images of single injection, the threshold of GFP intensities was determined as a level that was sufficient to remove the noise, but was able to conserve the shape of GC dendrites. As a result, even though the absolute threshold intensities were varied, the threshold in the histogram-adjusted images was all in the similar level at around 30%. By using the thresholded images, the percentages of GFP-positive glomeruli among all glomeruli were calculated.

In the double injection images, mean intensities of tdT and GFP within individual glomeruli were measured and were normalized by the median intensities in a single image of $211.7 \times 211.7$ µm$^2$. The ratios ($R_{tdT/G}$) of the normalized mean intensities of tdT to those of GFP ($D_{tdT}/D_{GFP}$) were then calculated, and the cumulative distributions of $R_{tdT/G}$ were plotted. To compare the distributions $R_{tdT/G}$ obtained from experimental data ($R_{tdT/G\text{-data}}$) with random configurations, we also calculated the ratios of randomly shuffled pairs of tdT and GFP intensities ($R_{tdT/G\text{-shuffle}}$) obtained from each image, because the random configuration could be equivalent to the case that any pairs of tdT- and GFP-positive dendrites are included in individual glomeruli. To achieve higher randomization, the shuffled pairs were generated 50 times. In addition to the comparison in cumulative distributions, we further compared the broadness of distributions between $R_{tdT/G\text{-data}}$ and $R_{tdT/G\text{-shuffle}}$ by subtracting standard deviation, $\hat{\sigma}$, which was estimated by fitting to a lognormal distribution. The broadness difference was calculated by the following formula:

$$\overline{\triangle\hat{\sigma}} = \frac{1}{n} \sum_{i=1}^{n} \left(\hat{\sigma}_{shuffle(i)} - \hat{\sigma}_{data}\right) \qquad (2)$$

where $n$ is the number of times (50) that shuffled pairs were generated. Since the variance of the ratio of two non-negative random variables ($D_{tdT}/D_{GFP}$) can be estimated approximately as the following equation[59],

$$\widehat{\sigma^2} \cong \frac{\sigma_{GFP}^2 \mu_{tdT}^2}{\mu_{GFP}^4} + \frac{\sigma_{tdT}^2}{\mu_{GFP}^2} - \frac{2Cov(D_{tdT}, D_{GFP})\mu_{tdT}}{\mu_{GFP}^3} \qquad (3)$$

where $\mu$ and $\sigma$ denote the mean and standard deviation, respectively, it is reasonable to expect the largest variance when $D_{tdT}$ and $D_{GFP}$ have perfect negative correlation. Thus, the positive correlation underlying the data is eventually represented by positive sign of broadness difference, considering that shuffling of pair minimizes the covariance. Whereas comparison between Pearson correlation coefficients cannot detect small differences clearly, demarcating the direction of correlations by their corresponding sign of broadness difference can be a useful approach.

**Analyses in the z-stack images.** For the analyses of tdT- and GFP-positive GC dendrites in terminals of PN-MFs and DCoN-MFs, the z-stack images were taken from slices obtained from mice that were subjected to double injection of AAV-GABRα6-tdT at P7 and AAV-GABRα6-GFP at P12 into the cerebellum together with injection of AAV-sSyn-mTagBFP into PN or DCoN at P12 (Fig. 2c). In these slices, mTagBFP was equally present in the specific MFs regardless of the depth in slices, unlike staining with antibodies of marker proteins of glomeruli, so that they were appropriate for the 3D analyses. Thin MF axons were filtered out from the z-stack images of mTagBFP, and the labeled MF terminals with a volume size of >25.5 μm³ were detected by 3D objective counter functions in ImageJ software (Fig. 2d). If MF terminal clusters or MF axons were included, they were manually removed. The intensities in deeper optical sections of tdT and GFP images were adjusted by the histogram equalization ("Equalize Intensity in Z" function in NIS software), and mean intensities of tdT ($I_{tdT}$) and GFP ($I_{GFP}$) in individual MF terminals were measured. Normalized mean intensities of tdT ($D_{tdT}$) and GFP ($D_{GFP}$) were then calculated as follows:

$$D_{tdT} = \frac{I_{tdT} - IB_{tdT}}{IW_{tdT}} \tag{4}$$

$$D_{GFP} = \frac{I_{GFP} - IB_{GFP}}{IW_{GFP}} \tag{5}$$

where $IB$ is background signals calculated in the PC layer or white matter in z-projection images, and $IW$ is mean intensities in whole GCL of z-projection images. Total numbers of MF terminals detected were 1277 for DCoN and 1191 for PN. Seven mice were used for each labeling of PN-MFs or DCoN-MFs, and three z-stack images (211.7 × 211.7 × ~30 μm³) of lobule IV/V per one mouse were used for the analyses. For comparison, $D_{tdT}$ and $D_{GFP}$ in non-specific glomeruli were detected by a method described above (see a section "Analysis of the synaptic connections of GC dendrites in the individual glomeruli"), and are shown in Fig. 2g.

For analyses of MF terminal volume, the z-stack images were taken from cerebellar slices obtained from mice that were subjected to injection of AAV-sSyn-GFP into PN or DCoN at P1 or P2. Individual MF terminals labeled with GFP were detected as described above, and averaged volume of these terminals in a single z-stack image was calculated. Four mice were used for each labeling of PN-MFs or DCoN-MFs at different ages, and three z-stack images (211.7 × 211.7 × ~30 μm³) of lobule IV/V per one mouse were used for the analyses. In case of P21, MF terminals used for the analyses of GC dendrites included were also used for the analyses of MF terminal volume.

**Construction of the hypothetical network model.** The network model was coded in Python and NetworkX library was utilized to visualize network structures in the model[60].

Our model network consists of 2 types of nodes, GCs and glomeruli (MF terminals), and these 2 types of nodes are connected via edges, GC dendrites. Because of previous reports estimating the 3:1 ratio of GCs to glomeruli[2] and showing mostly 4 GC dendrites[5,61], 3000 GCs and 1000 glomeruli were considered in our model, and the degree of a GC node was set to be 4. The GCs had ID numbers, which reflect the sequence of PFs, and 4 GC dendrites connecting to the same GC nodes have the same ID numbers. The random connection was first constructed in a way that each GC dendrite was randomly placed on one of glomerulus nodes with a probability of discrete uniform distribution, within a minimal constraint that GC dendrites originating from the same GC nodes were not connected to the same glomerulus nodes[2]. For the simplification of analyses on the synaptic connections from MF terminals to GC dendrites according to the PF locations, we used this simple approach to make network model, which does not include morphological and geometrical constraints in GCL. Nevertheless, the degree of glomerulus nodes in our model is similar to the results of model including these constraints[2], in terms of that the degree of glomerulus nodes was 12 in average and followed binomial distribution (Fig. 4a). Further, considering our results showing that GC somas were randomly distributed regardless their PF locations (Fig. 1c–e), our simple network model is valid for the analyses of PF location-related synaptic connections.

The preferential and avoidance connections were generated by rewiring the random connections in a way that the averaged proximity of ID numbers of GC dendrites within glomerulus nodes was increased for the preferential connections and decreased for the avoidance connections. To minimize the alteration, the degree of individual glomerulus nodes was conserved, and rewiring was performed by the permutation of a randomly picked pair of GC dendrites (Supplementary Fig. 6). The two GC dendrites had different ID numbers, represented as $x_p$ and $y_q$,

and were on the different glomerulus nodes, named X and Y. The execution of permutation was determined based on 'the score function', $S_0$ or $S$, which was sum of averaged ID number distances ($D_x + D_y$ or $d_x + d_y$) between picked and other GC dendrites in the glomerulus nodes. The $S_0$ represents before and the $S$ represents after the hypothetical permutation. The $S_0$ and $S$ are formulated as follows:

$$S_0 = D_x + D_y = \left[\frac{1}{N-1}\sum_{k=1}^{N}|x_p - x_k|\right] + \left[\frac{1}{M-1}\sum_{k=1}^{M}|y_q - y_k|\right] \tag{6}$$

$$S = d_x + d_y = \frac{1}{N-1}\left[\left(\sum_{k=1}^{N}|y_q - x_k|\right) - |y_q - x_p|\right] + \frac{1}{M-1}\left[\left(\sum_{k=1}^{M}|x_p - y_k|\right) - |x_p - y_q|\right] \tag{7}$$

where $x_k$ and $y_k$ are ID numbers of other GC dendrites in the glomerulus nodes, and $N$ and $M$ are degrees of the glomerulus nodes. For the preferential connection, permutation was executed if $S$ was smaller than $S_0$, while it was executed if $S$ was larger than $S_0$ for the avoidance connection. By varying the repeats of permutation trials, the level of preferential or avoidance connections could be manipulable, and three levels of connections, mild, moderate, and severe connections, are presented in this study (2500, 5000, and 7500 repeats for the preferential connections; 1000, 4000 and 10,000 repeats for the avoidance connections; Fig. 5c). Individual connection models have 50 different configurations, and three levels of preferential and avoidance connection models were made from 50 different random connection models.

**Confirmation of the basic model properties.** As mentioned above, preferential and avoidance connections were made with the intention of making the proximity of GC dendrite ID numbers within glomerulus nodes increased and decreased, respectively. To confirm this, we calculated PI as the inverse of average pairwise differences ($\bar{\mu}_d$) among all GC dendrite ID numbers in individual glomerulus nodes. The PI in node X ($\Phi_X$) with the degree of $N$ is formulated by the following equation:

$$\Phi_X = \frac{1}{\bar{\mu}_d} = \frac{N(N-1)/2}{\sum_{i=1}^{N-1}\sum_{j=i+1}^{N}|x_j - x_i|} \tag{8}$$

where $x_i$ and $x_j$ denote the ID numbers of GC dendrites in the glomerulus node. The PI in all glomerulus nodes was then averaged to compare the PI among different connection models. As we intended, PI was increased in the preferential connection and decreased in the avoidance connection (Fig. 4b).

The experimental results of single labeling by the injection of AAV-GABRα6 demonstrated that more than 80% of glomeruli included GFP-positive GC dendrites (Fig. 4c). We tested if our network model could reproduce similar results. Considering that a single injection of AAV-GABRα6 resulted in the labeling of approximately 30% of GCs (Fig. 4c), whose PFs were bundled, we looked at GC dendrites connecting to 30% of GC nodes with consecutive ID numbers, and percentages of glomerulus nodes including these GC dendrites were calculated in the model. The results showed that more than 80% glomerulus nodes had these GC dendrites in any connections of network models, confirming the experimental observation (Fig. 4d).

**Interpretation of the connections of GC dendrites in the glomeruli by the network model.** By utilizing the network model described above, we estimated how the distributions of $R_{tdT/G}$, the ratios of two groups of GC dendrites within individual glomeruli, could be varied according to the types of connections, random, preferential, or avoidance connections. The labeling of two groups of GCs by the double injection of AAV-GABRα6 was mimicked by assigning red and green colors to two groups of GC nodes in the model network, corresponding to the expression of tdT and GFP in GCs. Consequently, dendrites connecting to these groups of GC nodes were assumed to include tdT or GFP. Both groups of GC nodes were basically 30% among all GCs (900 GC nodes), to mimic labeling in experiments. The numbers of tdT ($N_{tdT}$)- and GFP ($N_{GFP}$)-positive dendrites were then counted in the glomerulus nodes. However, in general, the fluorescent intensities detected in experiments were highly diverse, and were not integer like numbers of dendrites. Thus, to imitate the variability of fluorescent intensities, the numbers of GC dendrites in the glomerulus nodes were represented as the sum of random numbers drawn from a lognormal distribution with a mean value of 1.0 and a standard deviation of 0.6. In addition, to simulate the background noise, we added local random noise derived from a lognormal distribution with a mean value of 0.3 and a standard deviation of 0.1 onto the both numbers of tdT- and GFP-positive dendrites. The ratios ($R_{tdT/G-model}$) of these numbers ($N_{tdT}/N_{GFP}$) were then calculated.

To compare with the experimental results, we attempted realistic labeling in the model networks, by considering overlapping expression of tdT and GFP (Fig. 4e, f). The injection time-dependent expression patterns were mimicked by uniformly shifting the ID numbers of the labeled group of GC nodes as followings: 1–900 for P7, 351–1250 for P8, 701–1600 for P9, 1051–1950 for P10, 1401–2200 for P11, 1751–2650 for P12, and 2101–3000 for P13. To take into account the percentages of double-labeled GCs calculated from the experimental results, equivalent percentages of tdT/GFP double-labeled GC nodes were assumed in the end of the

former group and the beginning of later group. At the same time, numbers of labeled GC nodes were adjusted to keep 30% each of tdT- and GFP-positive GC nodes. In addition to $R_{tdT/G\text{-model}}$ obtained from the network models, the ratios of randomly shuffled pairs of tdT- and GFP-positive dendrite numbers ($R_{tdT/G\text{-shuffle}}$) were also calculated as was done with experimental data. The shuffled pairs were generated 20 times from data in one configuration. The cumulative distributions and broadness of distributions estimated from standard deviations were then compared between $R_{tdT/G\text{-model}}$ and $R_{tdT/G\text{-shuffle}}$. Since individual connection models have 50 different configurations, the differences in the broadness in the network model were formulated as follows:

$$\overline{\triangle \hat{\sigma}} = \hat{\sigma}_{shuffle} - \hat{\sigma}_{model} = \frac{1}{n \cdot m} \sum_{j=1}^{m} \sum_{i=1}^{n} \left( \hat{\sigma}_{shuffle(i)formodel(j)} - \hat{\sigma}_{model(j)} \right) \quad (9)$$

where n is the number of times (20) that shuffled pairs were generated, and m is the number of network configurations (50).

To further characterize properties of synaptic connections, distinct labeling was performed in two groups of GC nodes located at either near (1–900 and 1051–1950) or far (1–900 and 2101–3000) distance (Fig. 5), instead of the realistic labeling including double-labeling. The cumulative distributions of $R_{tdT/G\text{-model}}$ were compared among different types of connections (Fig. 5b, c). For the estimation of fine network structures in the mild preferential connection (Supplementary Fig. 7), groups of GC nodes including tdT or GFP were considered to be 5% (150 GC nodes) or 15% (450 GC nodes), and distinct labeling was performed in two groups of GC nodes located at either near (1–150 and 451–600 for 5%; 1–450 and 751–1200 for 15%), middle-range (1–150 and 1601–1750 for 5%; 1–450 and 1651–2100 for 15%) or far (1–150 and 2851–3000 for 5%; 1–450 and 2551–3000 for 15%) distance. The cumulative distributions and distribution broadness ($\hat{\sigma}$) of $R_{tdT/G\text{-model}}$ were compared between random and mild preferential connections.

**Model prediction of the connections of GC dendrites to specific MF terminals**. For the prediction of synaptic connections of GC dendrites to specific MF terminals, we considered glomeruli in the model as MF terminals. MF terminals in the mild preferential connection model were divided into populations by the following three different criteria: random, dominancy-dependent, or order-dependent categorization (Fig. 6a). For random categorization, MF terminals were randomly divided into three populations. For dominancy-dependent and order-dependent categorization, GCs with ID numbers of 1–1000, 1001–2000, and 2001–3000 were considered as D-GCs, M-GCs, and S-GCs, respectively, and MF terminals were divided according to the most dominant type or decreasing order of GC dendrites connecting to the MF terminals (Fig. 6a, Supplementary Fig. 8). To detect the most dominant type of GC dendrites, numbers of D-GC, M-GC, and S-GC dendrites within individual MF terminals were presented by numerical sequences of their ratios among all dendrites on the MF terminals (Supplementary Fig. 8a), and the sequences were arranged in descending order of ratios of either D-GC, M-GC, or S-GC dendrites (Supplementary Fig. 8b). The top 33% of MF terminals were considered as terminals that dominantly connect to that particular type of GC dendrites. As a result, three populations of MF terminals, D-GC-, M-GC-, and S-GC-dominant terminals, were obtained. To present the decreasing order of GC dendrites connecting to MF terminals, numbers of D-GC, M-GC, and S-GC dendrites on the MF terminals were ranked from the most abundant (0) to the fewest (2) GC dendrites (Supplementary Fig. 8a), and sequences of the rank of D-GC, M-GC, and S-GC were made (Supplementary Fig. 8c). For example, a MF terminal with {1, 2, 0} meant that the MF terminal connected most abundantly to S-GC dendrites and least abundantly to M-GC dendrites (S > D > M MF terminals). The order-dependent categorization resulted in six populations of MF terminals. We then assumed realistic labeling of GCs with tdT at P7 and GFP at P12, as described above, and counted the number of tdT- and GFP-positive GC dendrites in individual MF terminals. The $N_{tdT}$ and $N_{GFP}$ in each population were averaged. The series of processes were repeated in 100 configurations of the mild preferential connection model, and the averaged results (Fig. 6b–d) were used to compare with experimental observations of $D_{tdT}$ and $D_{GFP}$ in the DCoN-MF and PN-MF terminals.

**Statistics and reproducibility**. Statistical differences were determined by two-way ANOVA followed by the Bonferroni test for approximately normally distributed data with similar variances. For the comparison between normalized intensities of tdT and GFP in non-specific glomeruli, and between labeled GC fractions by single and second AAV injection, Mann–Whitney test was used to determine a statistical difference. The paired Student's $t$-test was used to compare EPSCs triggered by P7-PF and P12-PF stimulation, and to compare $R_{tdT/G}$ distribution broadness ($\hat{\sigma}$) between random and mild preferential connections with 5% or 15% GC labeling. Under the hypothesis that the broadness of the distribution in $R_{tdT/G}$ is equivalent to that in ratios of shuffled data, the difference in broadness ($\triangle \hat{\sigma}$) is supposed to be equal to 0. To test whether the broadness was different from the shuffled data, we therefore, used the one-sample Wilcoxon signed rank test and compared the differences in broadness ($\triangle \hat{\sigma}$) with test median 0. Sample sizes were not statistically predetermined but conform to similar studies, and are described in figure legends. The experiments were not randomized. The investigator was blinded for all analysis and quantifications. Most of experimental data are presented as mean ± s.e.m. Data for distances between the labeled PF bundles and PC somas are presented as boxplots with gray open circles representing individual data points, center lines

denoting the median, the lower and the upper bounds of the box corresponding to the 25th and 75th percentiles, respectively, and the whiskers denoting the minimum and maximum values. All experimentally obtained numerical data are provided in Supplementary Data 1. The statistical analyses were performed using GraphPad Prism or OriginPro software.

**Reporting summary**. Further information on research design is available in the Nature Portfolio Reporting Summary linked to this article.

## Data availability
The datasets generated and/or analyzed during the current study are available from the corresponding author upon reasonable request.

## Material availability
Plasmids generated for this study will be shared upon reasonable request.

## Code availability
MATLAB and python scripts used for analysis and modeling are publicly available at https://github.com/lab-taco/MF_GC_organization.

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

## Acknowledgements

We thank Dr. Jinhyun Kim and members of the Tanaka-Yamamoto laboratory for valuable discussions during the project. This work was supported by the Korea Institute of Science and Technology Institutional Program (Project No., 2E32211), the National Research Foundation of Korea (NRF) grant funded by the Korean Ministry of Science and ICT (MSIT) (NRF grant Nos., 2021R1A2C3009991, 2021R1C1C2007843, and 2022R1A2C2006857), and the National R&D Program through the NRF funded by MSIT (Grant No. 2021M3F3A2A01037811).

## Author contributions

T.K., K.T.-Y., and Y.Y. contributed to the conception and design of the study. T.K., H.P., K.T.-Y., and Y.Y. performed experiments, and imaging analyses. T.K. built computational models and performed simulations. T.K., H.P., K.T.-Y., and Y.Y. contributed to writing the paper.

## Competing interests

The authors declare no competing interests.
