## [Peer Review File · Communications Biology]

Reviewers' comments:

Reviewer #1 (Remarks to the Author):

In their manuscript entitled "Developmental timing-dependent organization of synaptic connections between mossy fibers and granule cells in the cerebellum", Taegon and colleagues studied whether the developmental maturation of granule cells is linked with specific circuit organization. This is an important question as cerebellar cortex receives a wide range of sensory and motor information. Sensory information inputs to the cerebellum include two principal pathways, the mossy fibers (MFs) and the climbing fibers. Mossy fibers convey sensory information from multiple origins (spinal cord, pontine nuclei, vestibular nerve and nuclei, lateral reticular nucleus, trigeminal and dorsal column nuclei). Sensory information from mossy fibers is then distributed to granule cells which directly excite Purkinje cells through the parallel fibers in the molecular layer. Here they focus on two main inputs, the pontine (PN) and the dorsal column nuclei (DCN), to study the specificity of the mossy fiber inputs on granule cell (GC) dendrites during development. They found that the connectivity between MF and GC matched the developmental timing of both partners. In addition, GCs that innervate the same sub-region of the molecular layer are preferentially connected by the same MFs. Although, the work support most of the conclusions, I have several concerns.

My major concerns are as follows:

-In figure-1B, the mean distance from the PC soma is not informative. The authors should plot the minimum and maximum distances from the Purkinje cell soma for each developmental point. This will give a much better view of the PF bundle organization. They can show the data as a box and whisker plots.

-The author should discuss the fact that for each injection point, the number of infected GCs is high and the bundle of PF in the molecular layer are quite wide which make it difficult to precisely link PF proximity, GC soma location and MF innervation. One way to argue might be to analyze in more details the results obtained in figure 3 with the double infection. Indeed, in this situation, only a small fraction of granule cells is labelled by GFP and tdT, and their PFs are restricted to a small fraction of the molecular layer.

-Do the double labelled GCs have their soma randomly distributed?

-Do they receive bias MFs connectivity when comparing P7-P9 and P10-P13 double labelled GCs, for example?

I think that, overall, the authors could extract more meaningful information with their dual-label data that will strengthen their conclusions and interpretation in their different models (preferential, random and avoidance).

In Figure 3A, the diagrams are confusing. For example, the author should consider showing only one MF that will contact one preferential GC dendrites organization, but not three identical MF that will contact different GC dendrites organization. Or, they can change the colors of the MFs to representing one specific color with one GC dendrites organization. The same apply to preferential, Random and avoidance panels.

Reviewer #2 (Remarks to the Author):

The authors present a detailed and careful study relating the birthdate of cerebellar granule cells to properties of their anatomical and synaptic connectivity patterns. While a number of the findings presented here have, as the authors comprehensively acknowledge, been demonstrated elsewhere, I feel that the number of new findings (dual-color synaptic connectivity comparisons, MF developmental analysis and labeling combined with granule cell subset labeling for connectivity comparisons), in addition to several methodological advantages of the current techniques compared to prior results, justify publication of the authors' manuscript.

The authors use a viral approach to label developmental-timing-restricted subsets of granule cells. Compared to similar prior studies, this technique helpfully seems to infect relatively more restricted subsets of granule cells, and thus to label relatively more narrow bands of granule cell axons in the molecular layer. The subsets also appear to be sized more consistently as the authors vary the injection date, which permits more careful subsequent quantitative comparisons by birthdate than possible in similar prior studies. These are all important advantages and enable the authors to probe anatomical details with more precision than previously possible. Consistent with prior results, they find that granule cell soma location is random, regardless of birthdate / PF location.

In addition, the viral strategy allows repeated dual color injections, thus the authors label two birthdate populations in individual animals. They find that later-born granule cells received relatively more pontine and dorsal column input. More significantly, this technique enables the authors to determine the frequency with which granule cell groups receive common or distinct synaptic inputs. They comprehensively varied the pair of birthdate-restricted-subsets, counted synaptic input overlaps, and compared these to computational models of synaptic wiring. In the end, they found a slight preference for birthdate-matched granule cells to wire to the same MF terminals.

On a superficial reading, it might seem that the connectivity data is not different from the prior study showing biased MF input to developmental-timing-restricted granule cell subsets. But the authors' key advantage is being able to examine multiple terminals at individual glomeruli. I.e., it is possible for a set of neurons to preferentially share a common source (nucleus) of input—without being any more likely to share input from the very same axons. Of course both factors will contribute to observed wiring patterns, but I'm curious whether the authors tried to tease apart this issue more specifically, given this advantage of their methodology.

Beyond this, I found the techniques to be very careful and precise, the analyses to be comprehensive, and the claims well-supported by the data.

Minor comments:

One thing I didn't see was the serotypes of the AAVs used. My understanding of multiple-injections is that there is some effect of prior infection on susceptibility to subsequent infection. In cases of completely nonoverlapping populations (very superficial vs very deep) this is moot, but for the overlapping populations, it might matter. I was curious whether the authors used the same serotype repeatedly in their dual-color injections, or found that it was effective to repeat the same serotype, and if they had any data or thoughts on the multiple-infection issue.

I'm not sure "DCN" is a great abbreviation for dorsal column nucleus, given its ongoing usage for deep cerebellar nuclei (especially with both being significant sources of mossy fiber input!)

There are a few subject/verb agreement issues throughout, pls double check.

Reviewer #3 (Remarks to the Author):

In this manuscript,

Kim et al use a novel viral approach based on the transient mRNA transcription of the GABRA6 subunit to demonstrate a timing-dependent organization of synaptic contacts between mossy fibers (MF) and deep and superficial granule cells (GCs). Also, they demonstrate a specific organization of GCs with

MFs originating from the pontine nucleus (PN-MFs) versus the one originating from the dorsal column nuclei (DCN-MFs). This set a novel interpretation of cerebellar architecture based on parallel fibers (PF) location that in turns is impacting the connectivity and timing development of GCs population. The paper is well written, the experiments are carefully conducted and the statistics are correctly performed; the novelty and the impact of the findings are of undoubtedly relevance. My main concern in the overall flow of the manuscript lies in the absence of electrophysiological experiments; this is an important point considering that these approach was included in previous publication from the same senior author. The editor and the authors surely agree with me that a work with such fundamental novelties require more than one approach to the problem to support the conclusions in a stronger manner.

As a consequence of that I list here few experimental that I personally consider of high priority to improve the impact of this novel insights.

1) The best scenario would be to take advantage of the Chr2-GFP version of currently used AAV-GABR α 6-GFP. Electrophysiological recording from Purkinje cells (PCs) at different time of injection and plot amplitude and rise time of the EPSCs for the different time of injection (so the distance of the parallel fiber bundle from the PCs somata would give the ultimate validation of the author's statement). If this is not technically achievable (upon explanation) the authors can use the very same approach described in Park et al. 2019 in Fig. 2 C-D-E placing a stim electrode in the labelled PFs bundle and record from PCs plotting the rise time of the recorded EPSCs for the postnatal day injection. If the hypothesis is correct, slower rise time are expected for later stage injection (PFs bundle further out from PCs somata).

2) It has been shown that GABR α 6 is expressed on both excitatory and inhibitory GCs synapses (Nusser et al. 1996). It is of great interest to see if the mRNA of the GABR α 6 is expressed in a timing dependent manner. The data clearly shows a strong decay in mRNA production after a certain time (\sim p21). Is it so for the protein as well? Did the author have performed any immunostaining for the GABR α 6 to check if is a similar decay is reflected on the protein level? In case the author have done already this validation in their previous works and I accidentally miss it, they can just better emphasize the results of this experiment in the text citing the source.

3) The manuscript reveals a series of differences in connectivity and topography between D-GCs and S-GCs. Are the authors expecting some differences in basic electrophysiological properties (resting membrane potential, excitability, passive properties?). Recordings from labelled D-GCs and S-GCs would definitively give a . This is a crucial step to dissect the degree of divergency between the two GCs group.

Minor:

1) In the conclusion, the section about the migration of GCs should be expanded with more literature in regards what instruct GCs to stop migrating and so making synaptic contacts

2) GCs have a different maturation time in a cerebellar region dependent manner (Brandalise et al. 2016, Legu e et al. 2016). The author should mention some of these works and speculate on how a different timing in PN-MFs and DCN-MFs synapses formation in different area of the cerebellum might be responsible for the different timing in GCs migration.

We would like to express our deep gratitude to all reviewers for supporting our paper and giving us important suggestions that certainly help us to improve our paper. We believe that we thoroughly addressed all comments raised by reviewers, and consequently our revised manuscript is much improved. Our responses to each of reviewers' comments are listed below in blue, and modifications we made in the revised manuscript are summarized *in italic*.

Reviewer #1 (Remarks to the Author):

In their manuscript entitled “Developmental timing-dependent organization of synaptic connections between mossy fibers and granule cells in the cerebellum”, Taegon and colleagues studied whether the developmental maturation of granule cells is linked with specific circuit organization. This is an important question as cerebellar cortex receives a wide range of sensory and motor information. Sensory information inputs to the cerebellum include two principal pathways, the mossy fibers (MFs) and the climbing fibers. Mossy fibers convey sensory information from multiple origins (spinal cord, pontine nuclei, vestibular nerve and nuclei, lateral reticular nucleus, trigeminal and dorsal column nuclei). Sensory information from mossy fibers is then distributed to granule cells which directly excite Purkinje cells through the parallel fibers in the molecular layer. Here they focus on two main inputs, the pontine (PN) and the dorsal column nuclei (DCN), to study the specificity of the mossy fiber inputs on granule cell (GC) dendrites during development. They found that the connectivity between MF and GC matched the developmental timing of both partners. In addition, GCs that innervate the same sub-region of the molecular layer are preferentially connected by the same MFs. Although, the work support most of the conclusions, I have several concerns.

We appreciate this reviewer for giving us constructive suggestions. According to the suggestions, we revised figures and added results of new analyses in the revised manuscript. Responses to individual concerns raised by this reviewer are described below.

My major concerns are as follows:

(1) In figure-1B, the mean distance from the PC soma is not informative. The authors should plot the minimum and maximum distances from the Purkinje cell soma for each developmental point. This will give a much better view of the PF bundle organization. They can show the data as a box and whisker plots.

We would like to thank this reviewer for giving us a good idea to present the topographic information of GFP-positive PFs. We followed the suggestion and showed data in Fig. 1b and Supplementary Fig. 1c as boxplots.

(2) The author should discuss the fact that for each injection point, the number of infected GCs is high and the bundle of PF in the molecular layer are quite wide which make it difficult to precisely link PF proximity, GC soma location and MF innervation. One way to argue might be to analyze in more details the results obtained in figure 3 with the double infection. Indeed, in this situation, only a small fraction of granule cells is labelled by GFP and tdT, and their PFs are restricted to a small fraction of the molecular layer.

We agree that this is an important issue to be considered. While our technique using AAV-GABR α 6 enables stable and handy labeling of GCs according to their PF locations, it is true that the resolution to dissect GC networks is not very high due to labeling in approximately 30% of GCs. To overcome such limitation, this reviewer's suggestion, i.e., utilizing the double-labeled GCs, is a good practical way for some analysis. Related to this comment, we added results of further analyses (please see details below), and described the limitation of our technique in the Discussion (p. 20, lines 485 – 492) of the revised manuscript.

(2)-1 Do the double labelled GCs have their soma randomly distributed?

We could clearly detect tdT/GFP double-labeled GC somas after P7&P9, P7&P10, or P10&P13 injection, so that we followed this reviewer's suggestion and analyzed their distributions by

using G, F, and Ripley's K functions. The additional analyses confirmed their random distributions. Although estimated K functions in large radius tend to be closer to clustered configurations, this is likely because Ripley's K function is not optimal for analyzing the distributions of very small numbers of particles in large areas. *In the revised manuscript, the results of analyses in double-labeled GC somas are shown in Supplementary Fig. 7a and are described in p. 14, lines 325 – 334.*

(2)-2 Do they receive bias MFs connectivity when comparing P7-P9 and P10-P13 double labelled GCs, for example?

We also think that the analysis of MF-GC connectivity at high resolution could be theoretically achieved by the observation of two differentially double-labeled GC dendrites in individual glomeruli. However, there are three challenges to overcome, for the implementation of such analysis.

- (i) The first challenge is the establishment of stable and appropriate labeling for the analysis of MF-GC connectivity. For this analysis, we labeled two groups of GCs with tdT and GFP by the double injection, and took ratios of tdT and GFP signals in individual glomeruli (Fig. 3). To analyze double-labeled GC dendrites, we need to have two double-labeled GCs in individual mice. In other words, if we want to analyze GC dendrites doubly labeled by P7&P9 injection and those doubly labeled by P10&P13 injection, we have to inject different colors of AAV-GABR α 6 four times in individual mice at P7, P9, P10, and P13. However, it is not possible, because we do not have four different colors of AAV. Alternatively, two differentially double-labeled GCs can be made by the triple injection at P7, P10 and P13. While the probability of having three successful injections into the same areas of lobule IV/V is low, we have done it and took images (please see images in Fig. R1, for reviewer's reference).
- (ii) The second challenge is to precisely detect two differentially double-labeled GC dendrites in individual glomeruli, without including intersecting dendrites. Because GC dendrites are, unlike GC somas, originally thin and are not constant in shape or in size, it is difficult to precisely separate double labeled-GC dendrites and intersecting dendrites by using an available confocal imaging and analysis method (please see example images in Fig. R1, for reviewer's reference).
- (iii) The third challenge is the identification of individual glomeruli. For the identification, we have made a semi-automatic segmentation program, so that we could analyze large numbers of glomeruli (772–1,121 for each timing of injection). To perform the segmentation using the program, we need to use two kinds of GCL images, one stained with a Kv4.2 antibody (GC dendrites) and another stained with antibodies of vGLUT1 (MF terminal) and vGAT (Golgi cell axons). If we label three groups of GCs with three colors, as mentioned in (i), we can have only one kind of staining, because we currently have imaging system with four colors. Consequently, we cannot use the semi-automatic segmentation program.

Unfortunately, at this moment, the combination of these challenges greatly impedes us to analyze connectivity by utilizing double-labeled GCs.

Instead, to estimate detailed network structures, we used our mild preferential connection model that is capable of reproducing experimental results when we analyzed large numbers of GCs (Fig. 4). We tested whether the biased synaptic connections, i.e., the wider distributions of $R_{tdT/G}$ in the mild preferential model than those in the random model, can be detected by labeling only 5% or 15% GCs, instead of 30% GCs. We obtained results suggesting that biased synaptic connections could be detectable between GCs having PFs located beyond a distance corresponding to one-fifth to one-fourth (600 to 750 in 3,000) of all PFs, but not between GCs having PFs located within that distance. In the revised manuscript, *the results are shown in Supplementary Fig. 7b-e and are described in p. 14, lines 334 – 345.*

Fig. R1: Images of cerebellar slices obtained from mice that were subjected to triple AAV injection, AAV-GABR α 6-tdT at P7 (red), AAV-GABR α 6-mTagBFP2 (mTB) at P10 (blue), and AAV-GABR α 6-GFP at P13 (green). Slices were stained with Kv4.2 antibody (gray or magenta). Several examples of glomeruli are enlarged at the bottom. Intersecting (white arrows) and double-labeled (yellow arrows) dendrites were assumed by considering shape or size of fluorescent signals. Arrowheads indicate dendrites that are hard to be assumed.

I think that, overall, the authors could extract more meaningful information with their dual-label data that will strengthen their conclusions and interpretation in their different models (preferential, random and avoidance).

We thank this reviewer, again, for giving us the important suggestion. As mentioned above, we added figures for two analyses, a paragraph to describe the results of analyses, and discussion about limitations on numbers of labeled GCs, based on the suggestion. We think that these revisions strengthened our conclusions and improved our paper, and hope that this reviewer feels the same way.

In Figure 3A, the diagrams are confusing. For example, the author should consider showing only one MF that will contact one preferential GC dendrites organization, but not three identical MF that will contact different GC dendrites organization. Or, they can change the colors of the MFs to representing one specific color with one GC dendrites organization. The same apply to preferential, Random and avoidance panels.

As suggested by this reviewer, we modified the diagram in Fig. 3a. Specifically, we modified the preferential connection to the way that one type of MFs shown in blue preferentially connect with S-GCs shown in green.

Reviewer #2 (Remarks to the Author):

The authors present a detailed and careful study relating the birthdate of cerebellar granule cells to properties of their anatomical and synaptic connectivity patterns. While a number of the findings presented here have, as the authors comprehensively acknowledge, been demonstrated elsewhere, I feel that the number of new findings (dual-color synaptic connectivity comparisons, MF developmental analysis and labeling combined with granule cell subset labeling for connectivity comparisons), in addition to several methodological advantages of the current techniques compared to prior results, justify publication of the authors' manuscript.

The authors use a viral approach to label developmental-timing-restricted subsets of granule cells. Compared to similar prior studies, this technique helpfully seems to infect relatively more restricted subsets of granule cells, and thus to label relatively more narrow bands of granule cell axons in the molecular layer. The subsets also appear to be sized more consistently as the authors vary the injection date, which permits more careful subsequent quantitative comparisons by birthdate than possible in similar prior studies. These are all important advantages and enable the authors to probe anatomical details with more precision than previously possible. Consistent with prior results, they find that granule cell soma location is random, regardless of birthdate / PF location.

In addition, the viral strategy allows repeated dual color injections, thus the authors label two birthdate populations in individual animals. They find that later-born granule cells received relatively more pontine and dorsal column input. More significantly, this technique enables the authors to determine the frequency with which granule cell groups receive common or distinct synaptic inputs. They comprehensively varied the pair of birthdate-restricted-subsets, counted synaptic input overlaps, and compared these to computational models of synaptic wiring. In the end, they found a slight preference for birthdate-matched granule cells to wire to the same MF terminals.

On a superficial reading, it might seem that the connectivity data is not different from the prior study showing biased MF input to developmental-timing-restricted granule cell subsets. But the authors' key advantage is being able to examine multiple terminals at individual glomeruli. I.e., it is possible for a set of neurons to preferentially share a common source (nucleus) of input—without being any more likely to share input from the very same axons. Of course both factors will contribute to observed wiring patterns, but I'm curious whether the authors tried to tease apart this issue more specifically, given this advantage of their methodology.

Beyond this, I found the techniques to be very careful and precise, the analyses to be comprehensive, and the claims well-supported by the data.

We would like to thank this reviewer for recognizing our technical advantages and new findings.

Regarding the concerns that this reviewer mentioned in the last paragraph, we agree that it is important to consider two possibilities: (i) a group of GCs tend to simply share MF inputs from a common nucleus, or (ii) a group of GCs tend to share MF inputs from very same axons. Since our observation of overall preferential connections was mild, we believe that the synaptic connections cannot be solely attributed to (ii), and that both factors would contribute to the connectivity, as also noted by this reviewer. Indeed, a previous study of in-vivo calcium imaging suggested that PFs clustered in space could be activated either by different MFs or by same MFs, both of which carry the same information (Wilms and Häusser, 2015m Nat. Commun.). Unfortunately, we could not structurally tease a part this issue, because an establishment of at least two additional analyses is necessary, an analysis to identify collateral MF axons and an analysis to interpret or quantify the contribution by (ii). We simply assume that reconstruction of labeled MFs and GCs in entire image areas is at least necessary, it is not yet possible to separate individual GC morphologies due to high expression rate (~30% GCs). Nevertheless, it would be interesting to establish these analysis methods in the near future, and to clarify the issue by taking advantage of our labeling technique.

In the discussion of our original manuscript, we mentioned a possibility that MFs from not only collaterals of single neurons, but also different neurons in a specific precerebellar nucleus are involved in the preferential connections with GCs having neighboring PFs. *In the discussion of revised manuscript, we added a sentence that further analyses are necessary to precisely clarify the preferential connections with collateral MFs or different MFs (p. 22, lines 541 – 543).*

Minor comments:

One thing I didn't see was the serotypes of the AAVs used. My understanding of multiple-injections is that there is some effect of prior infection on susceptibility to subsequent infection. In cases of completely nonoverlapping populations (very superficial vs very deep) this is moot, but for the overlapping populations, it might matter. I was curious whether the authors used the same serotype repeatedly in their dual-color injections, or found that it was effective to repeat the same serotype, and if they had any data or thoughts on the multiple-infection issue. I'm not sure "DCN" is a great abbreviation for dorsal column nucleus, given its ongoing usage for deep cerebellar nuclei (especially with both being significant sources of mossy fiber input!) There are a few subject/verb agreement issues throughout, pls double check.

Thank you for pointing out the missing information. We used serotype 1 of AAV for all experiments in this study, and *included this information in the Materials and methods of revised manuscript (p. 24, line 573).*

We have not tested which combinations of serotypes are more effective for double labeling. However, when we compared the percentages of labeled GCs after a single injection versus a second injection of double injection, we discovered that the second injection had the equivalent labeling efficiency as the single injection at all times. The results indicate that prior injection with serotype 1 AAV had a minor impact on the subsequent injection in our experiments. *We added these results in Supplementary Fig. 4d, and mentioned it in the Results (p. 9 – 10, lines 213 – 219).*

Regarding the abbreviation of dorsal column nuclei, *we changed it to "DCoN" in the revised manuscript.*

There were subject-verb agreement errors in the original manuscript. We thank this reviewer for pointing them out. *We carefully checked throughout the manuscript and fixed them.*

Reviewer #3 (Remarks to the Author):

In this manuscript, Kim et al use a novel viral approach based on the transient mRNA transcription of the GABR α 6 subunit to demonstrate a timing-dependent organization of synaptic contacts between mossy fibers (MF) and deep and superficial granule cells (GCs). Also, they demonstrate a specific organization of GCs with MFs originating from the pontine nucleus (PN-MFs) versus the one originating from the dorsal column nuclei (DCN-MFs). This set a novel interpretation of cerebellar architecture based on parallel fibers (PF) location that in turns is impacting the connectivity and timing development of GCs population. The paper is well written, the experiments are carefully conducted and the statistics are correctly performed; the novelty and the impact of the findings are of undoubtedly relevance. My main concern in the overall flow of the manuscript lies in the absence of electrophysiological experiments; this is an important point considering that these approach was included in previous publication from the same senior author. The editor and the authors surely agree with me that a work with such fundamental novelties require more than one approach to the problem to support the conclusions in a stronger manner.

As a consequence of that I list here few experimental that I personally consider of high priority to improve the impact of this novel insights.

We would like to thank this reviewer for finding the advantages of our study, and for giving us suggestions to improve the impact of the novelties of our study. Even though some of suggested experiments are difficult for us to perform, we followed suggestions or came up with ideas to address all comments raised by this reviewer. As a result, the revised manuscript includes broader range of approaches, not only imaging and computational analyses, but also electrophysiological and western blotting analyses. Responses to individual comments raised by this reviewer are described below.

1) The best scenario would be to take advantage of the Chr2-GFP version of currently used AAV-GABR α 6-GFP. Electrophysiological recording from Purkinje cells (PCs) at different time of injection and plot amplitude and rise time of the EPSCs for the different time of injection (so the distance of the parallel fiber bundle from the PCs somata would give the ultimate validation of the author's statement). If this is not technically achievable (upon explanation) the authors can use the very same approach described in Park et al. 2019 in Fig. 2 C-D-E placing a stim electrode in the labelled PFs bundle and record from PCs plotting the rise time of the recorded EPSCs for the postnatal day injection. If the hypothesis is correct, slower rise time are expected for later stage injection (PFs bundle further out from PCs somata).

We agree that the optogenetic experiment using AAV-GABR α 6-mediated expression of Chr2 in GCs is a good validation of the location of labeled PFs. We have indeed used AAV-GABR α to express CatCh, a Chr2 mutant with an enhanced Ca²⁺ permeability, and succeeded to record photostimulation-evoked EPSCs (Kim et al., 2015, Brain Res.). Unfortunately, this experiment is not appropriate for the purpose of validation of the location of labeled PFs. To sufficiently evoke EPSCs, we had to apply blue light for relatively long time (> 100 ms), even though we used CatCh, which can trigger highly light-sensitive and fast neuronal activation, instead of Chr2. We assume that this is probably due to the low expression of CatCh triggered by AAV-GABR α 6. The AAV-GABR α 6 is a useful tool for selective labeling of GCs having PF bundles in a certain sublayer of ML, because it triggers sufficiently visible expression of fluorescent proteins, as seen in the current study. However, it does not always trigger strong expression of molecules, particularly larger molecules than fluorescent proteins. An example is the expression of GFP-fused tetanus toxin (TeTx) (Park et al., 2019, Cell Rep.). We could not detect any GFP signals, although we could confirm the TeTx expression by detecting the cleavage of VAMP2. In the optogenetic experiment, individual PFs appeared to be gradually activated during the long photostimulation, and consequently it took 50-100 ms to reach approximately 100 pA of peak EPSCs (Kim et al., 2015, Brain Res., please see Fig. R2, for reviewer's reference). Therefore, this experiment is not appropriate to measure rise time of EPSCs, which is the indicator of locations of stimulated PFs.

Fig. R2: The photostimulation-evoked EPSCs recorded from PCs. Five different traces are shown (modified from Kim et al., 2015, Brain Res.)

Instead of the optogenetic experiment, we followed this reviewer's second suggestion and performed experiments using electrical stimulation. We injected AAV-GABR α 6-tdT and -GFP at P7 and P12, respectively, and recorded EPSCs upon stimulation via the electrode placed on tdT- and GFP-positive PFs in slices obtained at P21-P25. As expected, the rise time of EPSCs evoked by the stimulation of GFP (P12)-PFs were significantly slower than that by the stimulation of tdT (P7)-PFs. *In the revised manuscript, we added the electrophysiological results in Supplementary Fig. 1d, e, and mentioned them in the Results (p. 5, lines 101 – 109).*

2) It has been shown that GABR α 6 is expressed on both excitatory and inhibitory GCs synapses (Nusser et al. 1996). It is of great interest to see if the mRNA of the GABR α 6 is expressed in a timing dependent manner. The data clearly shows a strong decay in mRNA production after a certain time (~p21). Is it so for the protein as well? Did the author have performed any immunostaining for the GABR α 6 to check if a similar decay is reflected on the protein level? In case the author have done already this validation in their previous works and I accidentally miss it, they can just better emphasize the results of this experiment in the text citing the source.

We thank this reviewer for giving us the suggestion of experiments that we have not addressed. Before explaining our results, we would like to confirm our understanding of properties of AAV-GABR α 6 that we used in the present study. The AAV-GABR α 6 is the AAV vector with the minimum region of GABR α 6 promoter, and we used it to express exogenous fluorescent proteins (e.g., AAV-GABR α 6-GFP). In general, it has been shown that AAV vectors with short form of promoters don't necessarily trigger the same temporal or spatial patterns of molecule expression with endogenous molecules that the promoters regulate (Nathanson et al., 2009, Front. Neural Circuits). It appears to be the case for AAV-GABR α 6, considering our results (A) and publications (B).

(A) Our results demonstrated that AAV-GABR α 6 triggered GFP or tdT expression when it is injected at P7 ~ P13, but not at after P21, indicating that AAV-GABR α 6 triggers expression of mRNA and protein of exogenous molecules in immature GCs.

(B) Expression of endogenous GABR α 6 mRNA was observed in mature GCs, but not in immature GCs (Zheng et al., 1993, Dev. Brain Res.; Varecka et al., 1994, J. Comp. Neurol.), and knock-in of lacZ under GABR α 6 promoter in mice resulted in expression of lacZ in mature GCs, but not immature GCs (Mellor et al., 1998, J. Neurosci.), indicating the endogenous GABR α 6 promoter is active in mature GCs.

Thus, the expression patterns of endogenous GABR α 6 and exogenous molecules induced by AAV-GABR α 6 appear to be distinct, which is common for AAV vectors with short promoters.

While revising, we realized that we had overlooked such distinct expression patterns in our original manuscript. Furthermore, despite the indications mentioned in (B), we could

not find literatures that clearly demonstrated the expression of endogenous GABR α 6 "protein" in mature GCs. We therefore performed western blotting (WB) and immunohistochemical (IHC) analyses of endogenous GABR α 6, as this reviewer suggested. The results showed that the expression of GABR α 6 protein increased gradually during postnatal development, and that it was found only in GCs in the GCL of developing and adult mice. Thus, we confirmed the expression of endogenous GABR α 6 "protein" in mature GCs.

In the revised manuscript, we described the expression of endogenous GABR α 6 in mature GCs (p. 6, lines 123 – 128), by referring publications and by showing our new results (Supplementary Fig. 2e, f). We also clarified the difference in expression patterns of endogenous GABR α 6 and exogeneous molecules induced by AAV-GABR α 6 (p. 6, lines 128 – 130).

3) The manuscript reveals a series of differences in connectivity and topography between D-GCs and S-GCs. Are the authors expecting some differences in basic electrophysiological properties (resting membrane potential, excitability, passive properties?). Recordings from labelled D-GCs and S-GCs would definitively give a . This is a crucial step to dissect the degree of divergency between the two GCs group.

We also think that functional analyses of different groups of GCs are important to understand the degree of their divergences. In fact, considering reported heterogeneity of overall GCs regarding not only electrophysiological properties (D'Angelo et al., J. Neurophys., 1998; Gandolfi et al., Front. Cell Neurosci., 2014; Dorgans et al., eLife, 2019; Straub et al., eLife 2020), but also gene expression (Lein et al., Nature 2007) and morphology (Houston et al., Sci. Rep., 2017), we expected some differences in these properties of GCs according to the locations of their PFs. On the other hand, because the heterogeneity in GCs was not very conspicuous, we also expected that the PF location-dependent differences would be subtle, if any. Thus, we thought that effective analyses from many GCs were required to detect the differences.

As this reviewer mentioned above, our group often uses patch clamp recording from Purkinje cells, which is more widely used than recording from GCs, but we don't have an experience of recording from GCs. Nevertheless, since we had a labeling technique using AAV-GABR α 6, we tried to record from labeled GCs. Unfortunately, we could not succeed effective recording. Therefore, to detect functional differences in different groups of GCs, we performed calcium imaging using an indicator dye in fresh cerebellar slices, and compared MF stimulation-dependent calcium responses between D-GCs, M-GCs, and S-GCs. These experiments allowed us to analyze functional properties from hundreds of GCs. We then found non-uniform calcium responses in GCs according to the location of their PFs, presumably arising from different distributions of NMDA and GABAA receptors at the synaptic and extrasynaptic regions. This study was published (Rhee et al., Mol. Brain, 2021).

Given these circumstances, we could not test differences in basic electrophysiological properties between D-GCs and S-GCs, yet we have found differences in functional properties by measuring calcium responses. *In the revised manuscript, we included a discussion highlighting the significance of conducting further analyses to clarify PF location-dependent differences, and referred our study of calcium imaging as an example (p. 20 – 21, lines 493 – 501).*

Minor:

1) In the conclusion, the section about the migration of GCs should be expanded with more literature in regards what instruct GCs to stop migrating and so making synaptic contacts.

As this reviewer suggested, we expanded the discussion of GC migration by referring two literatures about cessation of GC migration (p. 19, lines 446 – 461).

2) GCs have a different maturation time in a cerebellar region dependent manner (Brandalise et al. 2016, Legué et al. 2016). The author should mention some of these works and speculate on how a different timing in PN-MFs and DCN-MFs synapses formation in different area of the cerebellum might be responsible for the different timing in GCs migration.

We thank this reviewer for introducing these literatures. We discussed in our original manuscript regarding the difference in synaptic connectivity between our results and a report from Shuster et al. (Proc Natl Acad Sci U S A, 2021). We analyzed in lobule IV/V and Shuster et al. analyzed in lobule VI. Legué et al. (2016) demonstrated that GC generations in the central zone, including lobule VI, were delayed relative to other lobules. Therefore, a possibility may be that the differences in synaptic connectivity result from different developmental time courses of GCs in lobule IV/V and VI. *In the revised manuscript, we added this possibility in the discussion by referring two literatures that this reviewer introduced (p. 21 – 22, lines 518 – 524).*

REVIEWERS' COMMENTS:

Reviewer #1 (Remarks to the Author):

The authors have satisfactorily responded to my comments. I think the revised version is significantly improved and reports a novel and significant set of result, improving our understanding of MF-GC synaptic connectivity during development.

Reviewer #3 (Remarks to the Author):

In their manuscript entitled "Developmental timing-dependent organization of synaptic connections between mossy fibers and granule cells in the cerebellum", the authors investigated whether can be correlated with a certain pattern of circuit formation. This is still a topic under debate due to the technical challenge and the sub-area variability in the cerebellum itself.

Sensory information inputs reaches the cerebellum with two main inputs, the mossy fibers (MFs) and the climbing fibers. Sensory information from mossy fibers is then distributed to granule cells which directly excite Purkinje cells through the parallel fibers in the molecular layer. The author nicely demonstrate a specific

organization of GCs with MFs originating from the pontine nucleus (PN-MFs) versus the one originating from the dorsal column nuclei (DCN-MFs). This paved the way for a novel interpretation of cerebellar architecture based on parallel fibers (PF) location that in turns is impacting the connectivity and timing development of GCs population.

The authors have carefully looked after all the listed points and have spent an admirable effort in improving the detailed description of the applied methodologies. This will surely benefit any other groups that want to replicate and investigate further the topic.

I consequently support the acceptance of this manuscript with no more revisions from my side

We truly appreciate all reviewers for carefully reading our revised manuscript and acknowledging its improvement. Because both reviewers did not raise further comments that we should incorporate into our manuscript, we did not revise our manuscript except for editorial requests.

Reviewer #1 (Remarks to the Author):

The authors have satisfactorily responded to my comments. I think the revised version is significantly improved and reports a novel and significant set of result, improving our understanding of MF-GC synaptic connectivity during development.

We would like to thank this reviewer for giving us constructive comments at the initial review process. We are glad to hear that our revision satisfied this reviewer.

Reviewer #3 (Remarks to the Author):

In their manuscript entitled “Developmental timing-dependent organization of synaptic connections between mossy fibers and granule cells in the cerebellum”, the authors investigated whether can be correlated with a certain patten of circuit formation. This is still a topic under debate due to the technical challenge and the sub-area variability in the cerebellum itself.

Sensory information inputs reaches the cerebellum with two main inputs, the mossy fibers (MFs) and the climbing fibers. Sensory information from mossy fibers is then distributed to granule cells which directly excite Purkinje cells through the parallel fibers in the molecular layer. The author nicely demonstrate a specific organization of GCs with MFs originating from the pontine nucleus (PN-MFs) versus the one originating from the dorsal column nuclei (DCN-MFs). This paved the way for a novel interpretation of cerebellar architecture based on parallel fibers (PF) location that in turns is impacting the connectivity and timing development of GCs population.

The authors have carefully looked after all the listed points and have spent an admirable effort in improving the detailed description of the applied methodologies. This will surely benefit any other groups that want to replicate and investigate further the topic.

I consequently support the acceptance of this manuscript with no more revisions from my side.

We appreciate this reviewer for viewing our paper as beneficial to other groups, and for supporting our manuscript. We are glad to hear that no revisions are necessary.